# Multimodal dynamic and unclonable anti-counterfeiting using robust diamond microparticles on heterogeneous substrate

Tongtong Zhang [1], Lingzhi Wang[1], Jing Wang[2], Zhongqiang Wang [3], Madhav Gupta[1], Xuyun Guo [4], Ye Zhu[4], Yau Chuen Yiu[1,5], Tony K. C. Hui[5], Yan Zhou [6], Can Li [1], Dangyuan Lei [7], Kwai Hei Li[8], Xinqiang Wang [3,9], Qi Wang [3] ✉, Lei Shao [2] ✉ & Zhiqin Chu [1,10] ✉

The growing prevalence of counterfeit products worldwide poses serious threats to economic security and human health. Developing advanced anti-counterfeiting materials with physical unclonable functions offers an attractive defense strategy. Here, we report multimodal, dynamic and unclonable anti-counterfeiting labels based on diamond microparticles containing silicon-vacancy centers. These chaotic microparticles are heterogeneously grown on silicon substrate by chemical vapor deposition, facilitating low-cost scalable fabrication. The intrinsically unclonable functions are introduced by the randomized features of each particle. The highly stable signals of photoluminescence from silicon-vacancy centers and light scattering from diamond microparticles can enable high-capacity optical encoding. Moreover, time-dependent encoding is achieved by modulating photoluminescence signals of silicon-vacancy centers via air oxidation. Exploiting the robustness of diamond, the developed labels exhibit ultrahigh stability in extreme application scenarios, including harsh chemical environments, high temperature, mechanical abrasion, and ultraviolet irradiation. Hence, our proposed system can be practically applied immediately as anti-counterfeiting labels in diverse fields.

Counterfeiting is a growing global problem. It is pervasive in the manufacturing of products ranging from daily consumer goods to medicines and high-tech products. It can seriously endanger financial safety and national security, and even poses a threat to human health[1–5]. Various anti-counterfeiting strategies[1–5], including watermarks, holograms, barcodes, and two-dimensional codes, have been developed in recent decades to prevent counterfeiting. However, most of these tags are fabricated by reproducible deterministic processes, making the encoded information vulnerable to third parties who possess the relevant expertise. There is an urgent need to develop

[1]Department of Electrical and Electronic Engineering, The University of Hong Kong, Pokfulam Road, Hong Kong, China. [2]State Key Laboratory of Optoelectronic Materials and Technologies, Guangdong Province Key Laboratory of Display Material and Technology, School of Electronics and Information Technology, Sun Yat-sen University, Guangzhou, China. [3]Dongguan Institute of Opto-Electronics, Peking University, Dongguan, China. [4]Department of Applied Physics, Research Institute for Smart Energy, The Hong Kong Polytechnic University, Hung Hom, Hong Kong, China. [5]Primemax Biotech Limited, Hong Kong, China. [6]School of Science and Engineering, The Chinese University of Hong Kong, Shenzhen, China. [7]Department of Material Science and Engineering, City University of Hong Kong, Hong Kong, China. [8]School of Microelectronics, Southern University of Science and Technology, Shenzhen, China. [9]State Key Laboratory for Mesoscopic Physics and Frontiers Science Center for Nano-optoelectronics, School of Physics, Peking University, Beijing, China. [10]School of Biomedical Sciences, The University of Hong Kong, Hong Kong, China. ✉e-mail: wangq@pku-ioe.cn; shaolei5@mail.sysu.edu.cn; zqchu@eee.hku.hk

unbreakable anti-counterfeiting tags to combat the growing global prevalence of counterfeiting. One promising approach that has recently attracted attention involves the exploitation of security labels with physical unclonable functions (PUFs)[5]. A PUF exploits intrinsic random variations introduced by a non-deterministic fabrication process to form a secret key, which is internal to the PUF and is not assigned by an outside source. The random variation is thus analogous to a unique fingerprint[5,6]. So far, various types of materials (e.g., metallic[7–11] and semiconducting[12–15] materials, photonic structures[16,17], DNA nanostructures[18–20], and hydrogels[21,22]) have been tried, and various fabrication methods (e.g., drop-casting[7,8], inkjet printing[13,23], and self-assembly[24,25]) have been developed to demonstrate anti-counterfeiting with PUFs.

Optical PUFs have attracted growing attention because of the inherent randomness, controllability, and diversity in tuning the optical features of luminescent building blocks in multiple dimensions, e.g., fluorescence color, intensity, and lifetime value[3–5]. For example, the luminescence of these building blocks (e.g., quantum dots[13–15], perovskites[26,27], polymer dots[28,29], upconversion nanoparticles (NPs)[30,31], plasmonic NPs[7–11], phosphors[32–35], and lanthanides[36,37]) can be modulated (excited, quenched, or tuned) by a range of external stimuli (light, heat, chemicals, and mechanical force), making them promising candidates in advanced cryptography. Despite their undoubted significance, most optical PUFs show unsatisfied stability in complex environments[11,29]. Furthermore, most of them are produced by wet chemical synthesis in solution[5], which might not be compatible with microelectronics and may have an unfavorable influence on their primary product functions[11]. The successful deployment of next-generation anti-counterfeiting technologies primarily requires the development of robust PUF labels.

In recent years, diamond materials have found copious applications in various areas due to their extraordinary features, such as exceptional chemical inertness, excellent mechanical strength, and high-temperature stability[38,39]. In particular, the various photoluminescent defect centers in diamond, e.g., silicon-vacancy (SiV) centers, have attracted considerable attention for multiple applications, due to their unique optical and spectroscopic properties[40–42]. Particularly, the location, distribution, and concentration of those atomic defects in the diamond lattice are random due to their stochastic formation processes[43]. At the same time, the surface terminations, shape, and geometrical dimensions of the host diamond crystals could be easily modulated by their synthesis methods (e.g., high-pressure high-temperature (HPHT)[44], chemical vapor deposition (CVD)[45] techniques) and other post-treatments (e.g., surface modifications[46], reactive ion etching[47], and air oxidation[48]). As a result, these luminescent color centers are prone to the surface properties, structures, and shapes of diamond particles, and it is impossible for them to be cloned or counterfeited accurately. In particular, the SiV center exhibits a naked-eye-invisible near-infrared (NIR) emission at around 737 nm[40], reducing the difficulty of distinguishing confidential from disturbance information[14]. Thus, the color centers with NIR emissions, like SiV centers in diamond, have great potential to serve as optical PUF labels.

With the development of CVD and controlled doping techniques, it is now feasible to grow large areas of high-quality diamond particles with different color centers on various substrates at a relatively low cost[43,49]. The position, size, and shape of the grown diamond particles are highly dependent on various growth parameters, such as seeding, nucleation, and growth procedures[45]. These are all non-deterministic processes (Fig. 1a), satisfying the critical requirement for fabricating PUF labels.

In this work, we report a scalable, robust, trustworthy, and intelligent anti-counterfeiting strategy, using our previously developed high-quality CVD-grown diamond microparticles on heterogeneous substrates[50]. The developed diamond PUFs exhibit excellent performance in respect of capacity, diversity, safety, manufacturability, robustness and compatibility (Fig. 1b, Supplementary Table 1 shows a detailed comparison of the most studied optical PUFs). The photoluminescence (PL) intensity of the SiV centers and the light scattering patterns (and spectra) of the diamond particles are employed to enable multidimensional optical encoding with a large capacity (Fig. 1c). Moreover, post air oxidation treatment of the diamond PUFs enables their dynamic encoding function based on the non-deterministic changing of SiV PL signals and/or packed patterns of the diamond microparticles. Furthermore, the fabricated diamond-based anti-counterfeiting labels have shown ultrastability under extreme conditions, including harsh chemical environments, high temperature, mechanical abrasion, and UV light irradiation. The multimodal, dynamic encoding capability, and superb robustness of the diamond PUFs indicate they have great potential for realizing unbreakable anti-counterfeiting applications.

## Results

### Fabrication of the diamond-PUF anti-counterfeiting labels

We have recently demonstrated the fabrication of large-scale, high-quality diamond microparticles with SiV centers on the Si substrate using salt-assisted air-oxidized (SAAO) HPHT nanodiamonds (NDs) as CVD seeds[50]. As shown in Fig. 1a, the SAAO NDs[51] were used as CVD seeds to grow diamond microparticles on Si substrate. The CVD process we used (Fig. 1a) involved a combination of various non-deterministic random, uncertain, and non-clonable processes, including the employment of unclonable seeds, the random placement of the seed particles on the substrate by spin coating, and the growth process of the seeds. The size, shape, and spatial distribution of the grown diamond microparticles on substrates were inherently random, making them promising candidates as anti-counterfeiting labels. At the same time, the CVD process can be seamlessly integrated with modern microelectromechanical systems (MEMS) and nanoelectromechanical systems (NEMS)[52–54], and can be mass-produced on wafers (Fig. 2a). The flexible variability of CVD growth parameters (e.g., gas composition and flow rate, microwave power, pressure, growth time and temperature, diamond seeds, and substrates) together with the wafer-scale production ability make the mass commercial customization of the diamond-PUF labels possible. Moreover, the one-step heterogeneous CVD growth manner of the diamond particles on the substrates (rather than direct dip/spin-coating pre-synthesized particles on substrates as PUF labels) greatly guarantees the strong adhesion of diamond particles to the silicon substrate, which further enables the excellent stability of our diamond-PUF label as a whole device.

The as-grown diamond microparticles were measured to have an average particle size of around 1.32 μm, and displayed precise crystalline facets in the scanning electron microscopy (SEM) images (Fig. 2b–d). Moreover, the grown diamond particles can cover the Si wafer from 0.146% to 100% (calculated in an area of 30 μm × 40 μm, Supplementary Fig. 1) in area with the coverage tunable by adjusting growth parameters, providing ample space to accommodate the coding capacity of the proposed diamond-PUF labels. The Raman and X-ray diffraction (XRD) results (Fig. 2e, f) indicate the pure crystalline nature of the grown diamond microparticles. At the same time, etching of the Si substrate during the CVD process would produce a significant amount of dopant Si, which could enter the diamond lattice to form the SiV centers with superior fluorescence properties (Fig. 2g, h). Notice that other candidates in diamond, e.g., nitrogen-vacancy (NV), germanium-vacancy (GeV), tin-vacancy (SnV) color centers, etc. could also be in-situ introduced during CVD process by utilizing suitable substrates or precursors[55,56], which might further increase the functionality of diamond PUFs in the future.

### Static encoding based on scattered light signals of diamond microparticles

In practical anti-counterfeiting applications, the diamond-PUF tags are expected to have the capability of being detected and read out quickly,

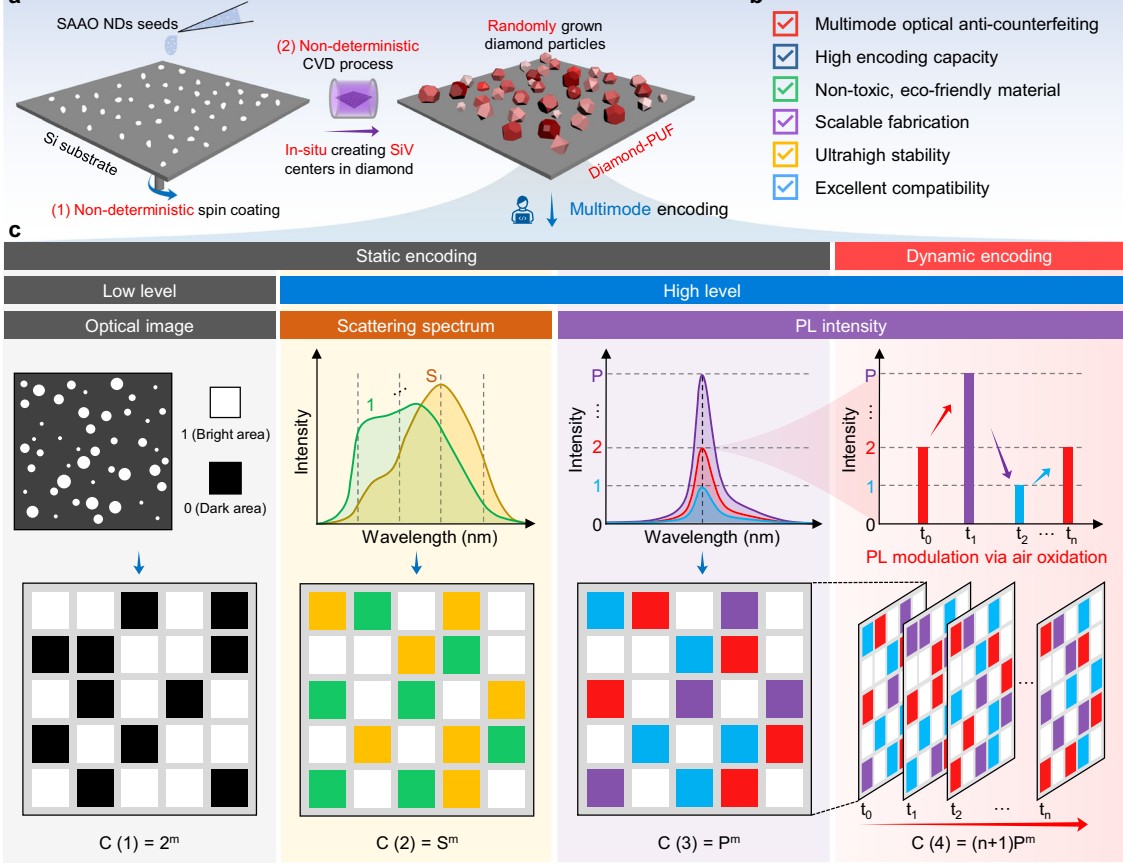

**Fig. 1 | The construction of diamond microparticles with multimodal and dynamic features for advanced anti-counterfeiting. a** Schematic illustration of the non-deterministic fabrication process of the diamond microparticles based anti-counterfeiting labels with physical unclonable functions (PUFs). **b** The favorable features of the developed diamond PUFs. **c** Representative multimode encoding process of the diamond-PUF labels. The bright-field optical imaging pattern, dark-field scattering spectrum, and photoluminescence (PL) intensity enable the multimode static encoding, and the time-dependent PL modulation via air oxidation enables the dynamic encoding.

cheaply and conveniently. Owing to the high refractive index (~2.4) of the diamond material[57], the fabricated diamond-PUF labels can be easily visualized by portable readout equipment, e.g., a smartphone-based microscope as illustrated in Fig. 3a-(i), enabling the diamond-PUF tags to be detected without needs for sophisticated instruments. Figure 3a-(ii) shows a typical optical image of the diamond-PUF label taken by the portable device. To generate quantitative metrics that are commonly used as figures of merit in PUF studies, the optical images are converted to security keys. For this purpose, the image is binarized and reduced to 50 pixels × 50 pixels (Fig. 3b, c). In Fig. 3c, white pixels correspond to bright regions which come from the light scattering signals of the diamond microparticles on the PUF label. Moreover, the Hamming distance (normalized), which measures the minimum number of substitutions between different rows within the corresponding binary encoding matrix, is shown in Supplementary Fig. 2a. The calculation result conforms to the normal distribution of the keys generated. This binarized image is then used to generate security keys 2500 bits in length, consisting of 1-bits and 0-bits. This procedure is repeated for 100 samples to generate 100 different security keys. For each of these keys, the uniformity (Supplementary Fig. 2b), similarity index (the percentage of the same number of pixels between two PUFs, Fig. 3d), and Hamming distance (Fig. 3e) are calculated. The calculated arithmetic average values of the uniformity (0.4996), similarity index (49.9978%), and Hamming distance (0.5000) are all close to the ideal values (i.e., 0.5, 50%, and 0.5), indicating the excellent uniformity, uniqueness, and randomness exhibited by the fabricated diamond

PUFs. In addition, our CVD diamond PUFs system is highly possible to incorporate various patterning techniques for realizing its practical applications. For example, the adopted Si substrate can be conveniently fabricated with patterns or makers, via well-established nanofabrication techniques, for the ease of locating the micro/nano diamond-PUF areas (Supplementary Fig. 3). Alternatively, patterned diamonds could be directly grown from pre-patterned CVD seeds[58], which offers a large space for product customization. To better demonstrate the anti-counterfeiting application, we further provide a detailed authentication protocol based on our diamond-PUF labels, as shown in Supplementary Fig. 4 and Supplementary Note 1. Moreover, our diamond-PUF labels are also suitable for other PUF-based applications, such as information encryption (Supplementary Fig. 5 and Supplementary Note 2).

Typically, the scattering pattern is insufficiently secure for applications pursuing high-level security, because it is relatively easy to replicate using 3D nanoprinting or other assembly technologies, though to do so admittedly requires both time and money. To improve the security of the PUF labels, the scattering spectra of the diamond particles can be further employed. The scattering spectra of diamond particles have been shown to be highly sensitive to topological structures such as size, shape, and crystallinity[59,60]. As a result, it is impractical to duplicate them with identical spectra accurately. Figure 3a (iii) and (iv) present typical dark-field microscopy and SEM images of a selected area of the diamond-PUF label. The diamond microparticles distribute randomly with bright scattering spots, and

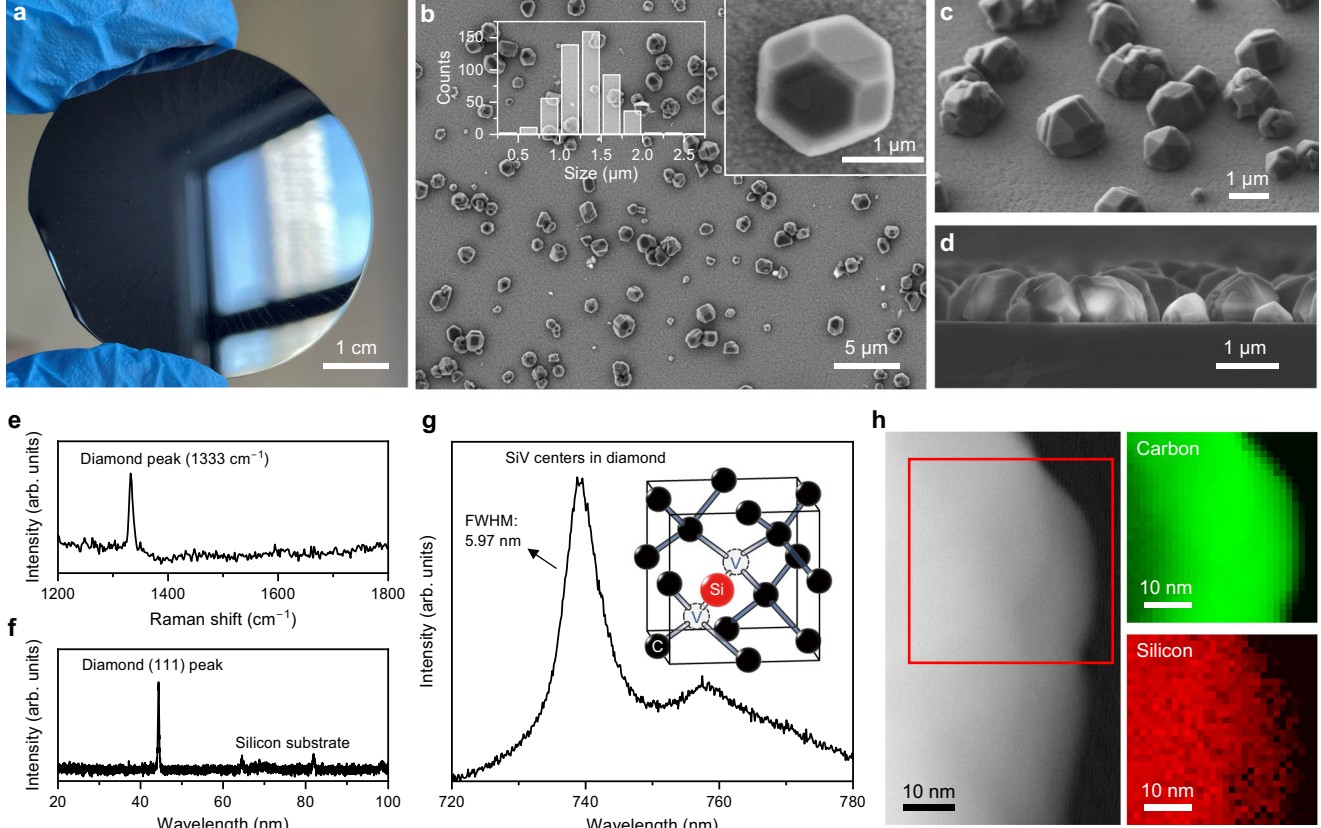

**Fig. 2 | Characterizations of the fabricated diamond microparticles on Si substrate. a** A photograph of the diamond PUFs fabricated on a 2-inch Si wafer. **b** Top-view scanning electron microscopy (SEM), **c** 45°-view SEM, and **d** cross-sectional SEM images of the fabricated diamond microparticles on the Si substrate. The particle size distribution of the diamond microparticles is shown in (**b**). **e** Raman, **f** X-ray diffraction (XRD), and **g** PL spectra of the fabricated diamond microparticles on the Si substrate. A representative atomistic diagram of a silicon-vacancy (SiV) center in a diamond unit cell is shown in (**g**). **h** Scanning transmission electron microscopy (STEM) image and corresponding electron energy loss spectroscopy (EELS) elemental maps from a diamond microparticle.

the detailed shape and corresponding scattering spectra of some particles are measured (Fig. 3f). The scattering peaks mainly range from 450 to 800 nm, and each spectrum has its unique peak position, intensity and spectral linewidth. At the same time, according to the SEM images, no particles share the same shape, i.e., each diamond microparticle is a unique specimen of its kind. Therefore, the scattering spectra of the diamond particles, together with their shape feature and spatial location relationships, could be employed as the fingerprints of the diamond PUFs. To explore the possibilities of modulating scattering spectra, we further performed a detailed finite-difference time-domain (FDTD) numerical simulation study of scattering spectra of diamond microparticles with different shapes (Fig. 3g and Supplementary Fig. 6a), heights/sizes (Fig. 3h and Supplementary Fig. 6b), substrates (Supplementary Fig. 6c), and crystallinity (Supplementary Fig. 6d). The results indicate that there is great potential for using the diamond scattering spectrum as a high-level anti-counterfeiting method. At the same time, because of the high refractive index of diamond, even if it is covered by some protection layers like $SiO_2$ or $Al_2O_3$, it will still have obvious light scattering signals for readout.

### Static encoding based on SiV PL intensity

Besides using light scattering signals from the diamond microparticles as an anti-counterfeiting method, it is also possible to achieve high-level encoding using the PL signals from the color centers (e.g., SiV centers) embedded in the diamond lattice. The fabricated diamond microparticles with a large amount of dopant Si atoms exhibit significant SiV signals (Fig. 2g, h). We encode the diamond-PUF label with the SiV PL intensity of each pixel in fluorescence image. As illustrated in Fig. 4a,

upon a 532 nm laser irradiation, the diamond-PUF label produces a unique optical fingerprint, i.e., a confocal fluorescence image with an area of $100\,\mu m \times 100\,\mu m$ (shown in Fig. 4b), which is impossible to imitate. The corresponding distribution of the PL intensity is shown in Supplementary Fig. 7. Due to the great diversity of the SiV centers in each diamond particle, we were able to achieve high-level (e.g., quaternary, decimal and hexadecimal) encoding of the PL intensity at each pixel. Figure 4c shows the quaternary, decimal, and hexadecimal encoding principle using the cumulative curve of the PL intensity (photon counts: counts per second, cps) in the measured area of Fig. 4b, enabling a uniform distribution of the generated keys. Figure 4d–f present the quaternary, decimal and hexadecimal PL encoding results of the measured area in Fig. 4b. It can be observed that the two measurements of the same label have a high similarity, enabling true and false labels to be easily distinguished. The distributions of the normalized inter- and intra-Hamming distances of the generated quaternary, decimal, and hexadecimal keys are shown in Fig. 4g. The results indicate the extraordinary ability of the current diamond-PUF system to generate distinct codes. At the same time, the influences of the measurement conditions (e.g., ambient light, exposure time and laser power) while capturing the PL images could not affect the final keys generated by the same diamond-PUF label based on such self-compared PL-based encoding process (Supplementary Fig. 8 shows the results).

### Dynamic encoding based on SiV PL intensity modulation via air oxidation

Recently, the modulation of SiV PL in diamonds has been demonstrated by several treatments, including thermal oxidation[61,62] and

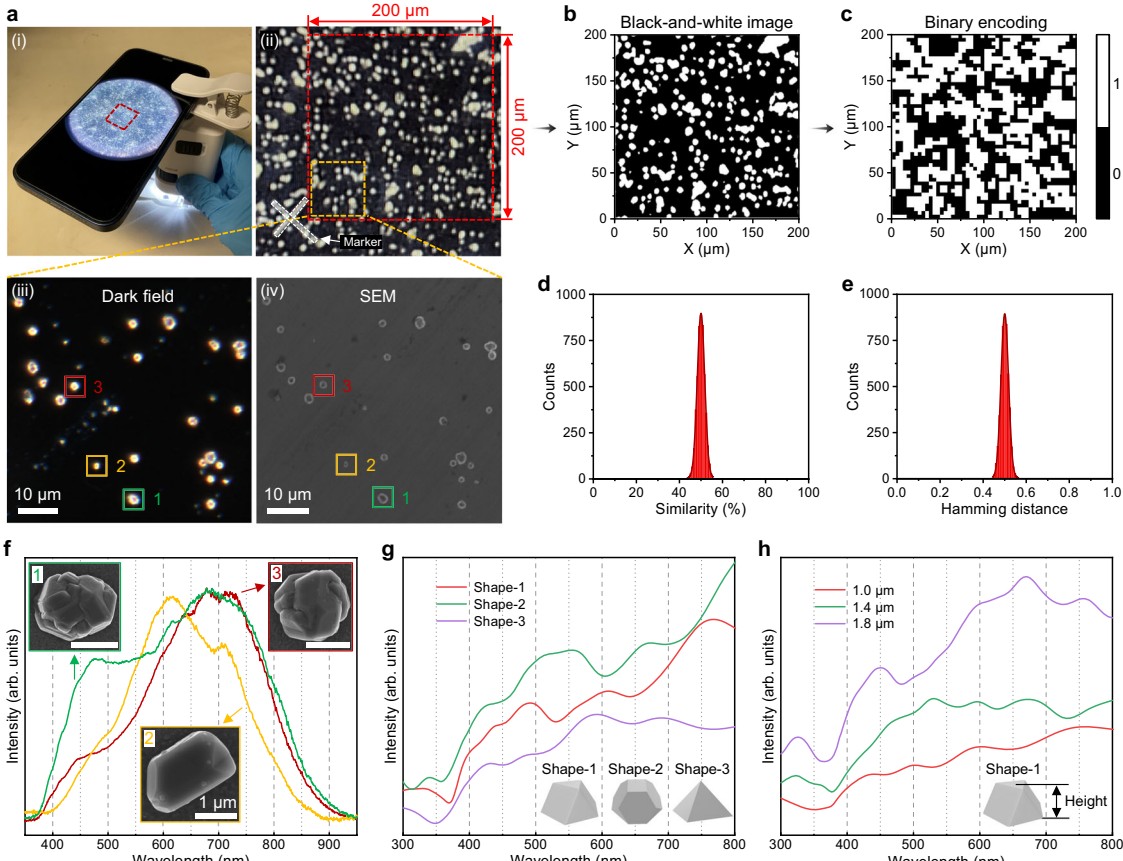

**Fig. 3 | Encoding based on scattered light signals of the diamond micro-particles. a** (i) An authentication setup comprising a portable microscope and a smartphone. (ii) An optical image of the diamond-PUF label taken by the portable microscope. (iii) Dark-field microscopy and (iv) SEM images of the yellow dashed area in (ii). **b** The black-and-white image is converted from the red dashed area of 200 μm × 200 μm in (**a**-(ii)). **c** The corresponding 2D binary encoding matrix of (**b**) based on dark (0) and bright (1) levels at each pixel, with a resolution of 50 pixels × 50 pixels. Distributions of **d** similarity and **e** Hamming distances (normalized) for keys generated from 100 labels. **f** The scattering spectra and corresponding SEM images of three random diamond microparticles in the selected area. Numerical calculated scattering spectra of diamond microparticles with different **g** shapes (~1 μm) and **h** heights/sizes (Shape-1: cubo-octahedral).

annealing[63]. This dynamic and controllable SiV PL system could potentially be a candidate for developing advanced time-dependent encoding method. As illustrated in Fig. 5a, the modulation of the SiV PL in the fabricated diamond microparticles is achieved by air oxidation (temperature: 600 °C, time interval: 15 min). Furthermore, the diamond microparticles in the selected area did not show any morphological change after four rounds of air oxidation treatment (Supplementary Fig. 9). Confocal fluorescence imaging of the red dashed area of 20 μm × 20 μm in Fig. 5a at three different times, $t_0$, $t_x$ (30 min), and $t_y$ (60 min), indicates an apparent change in the measured PL counts at each particle (Fig. 5b). The PL intensity variation differs from particle to particle (Fig. 5c). The measured confocal fluorescence images at different time (Fig. 5b) can therefore be digitized into different codes (Fig. 5d) which are not strongly correlated. The similarity (Fig. 5e) and Hamming distances (Fig. 5f) analysis of the decimal keys generated at the same and different time conform to the success of time-dependent encoding based on PL intensity modulation via air oxidation. For the diamond-PUF chip at the same time, the similarity index and Hamming distances are centered at 72.80% and 0.27, with standard deviations of 9.38% and 0.09, respectively. If they are from a different time, the similarity index and Hamming distances have means of 25.69% and 0.743, with a standard deviation of 1.24% and 0.01. Therefore, air oxidation on the diamond-PUF results in the overall change of the SiV PL intensity of each particle, in response to the construction of the dynamic diamond-PUF keys. In this situation, new keys will be generated after

each round of air oxidation due to the random modulation of SiV centers in diamond during oxidation. This non-deterministic time-dependent encoding strategy allows the encoded keys of the same PUF label to be personally changed when the original label faces malicious attacks and the risk of being duplicated, a desirable feature in the design and fabrication of anti-counterfeiting material with high-level security.

Furthermore, the packed patterns of the diamond microparticles on the substrate can be dynamically controlled by air oxidation (Supplementary Fig. 10). The multigenerational microstructures of the diamond microparticles could also work as PUF patterns to facilitate time-dependent encoding. Therefore, it is possible for end users of the diamond-PUF anti-counterfeiting system (Fig. 5g) to use the strategy we have developed to construct their own databases for a higher level of security.

**The stability of the diamond-PUF anti-counterfeiting labels**
The current diamond microparticles on the silicon substrate have delivered excellent performance in generating PUFs. For practical applications, the chemical, mechanical, thermal, and photo-stabilities of the anti-counterfeiting tags are also essential. We have therefore characterized the superior robustness of the diamond PUFs (Fig. 6) towards alkali (NaOH), acid (HCl), salt (NaCl), mechanical force, high temperature (400 °C), and UV light. Qualified anti-counterfeiting tags should remain stable even after exposure to these extreme conditions. No apparent change in optical response (Fig. 6a, d) and morphology

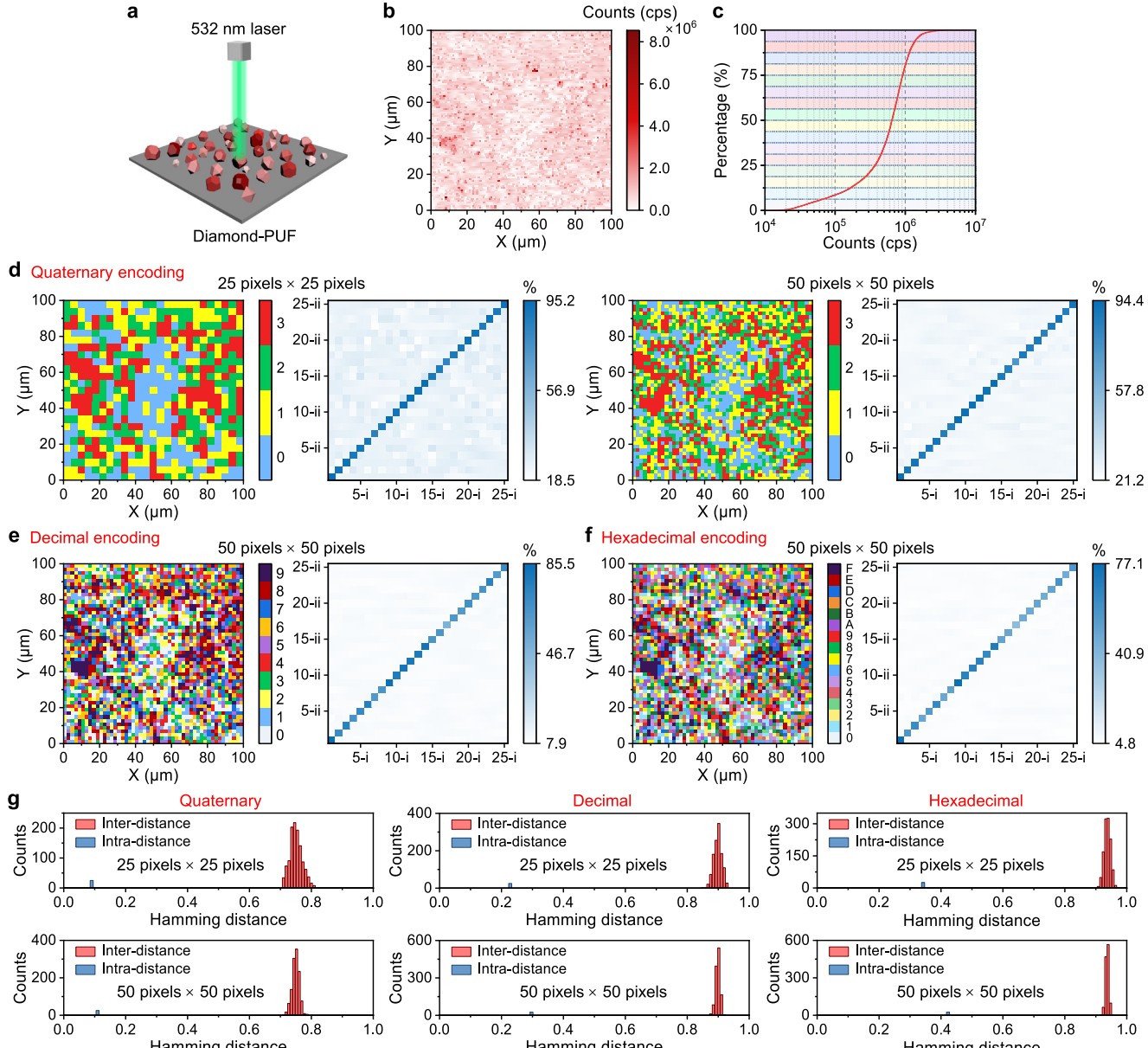

**Fig. 4 | Static encoding based on PL intensity of SiV centers in diamond microparticles. a** Illustration of the readout process of the diamond-PUF label using a 532 nm green laser. **b** A confocal fluorescence image of the diamond-PUF label with an area of 100 μm × 100 μm. **c** The cumulative curve of the PL intensity (photon counts: counts per second, cps) in the measured area shown in (**b**). **d** The digitization of the quaternary encoding of the PL intensity levels at each pixel at a resolution of 25 pixels × 25 pixels and 50 pixels × 50 pixels. The digitization of **e** decimal encoding, and **f** hexadecimal encoding at a resolution of 50 pixels × 50 pixels. The pairwise match of 25 PUF labels with quaternary, decimal and hexadecimal encodings of PL intensity levels at each pixel are shown next to the corresponding encoding matrix. The x-axis and the y-axis represent the first and the second measurement of the labels, respectively, and the color bar shows the similarity index. **g** The corresponding Hamming inter- and intra-distance (normalized) of the quaternary, decimal and hexadecimal keys generated from 25 labels.

(Supplementary Figs. 11 and 12) of the diamond microparticles was observed after this series of extreme treatments, indicating their excellent stability in different extreme applications scenarios. Moreover, the similarity (Fig. 6b) and Hamming distances (Fig. 6c) analysis of the binary keys generated before and after the treatments conform to the encoding capability of the diamond PUFs. Furthermore, we examined the photostability of the SiV centers in the diamond PUFs by irradiating them with a 532 nm laser for 2 h, and they displayed an excellent photostability deviation of less than 1.19% during the measurement. These results demonstrate that the proposed diamond PUFs possess remarkable chemical, mechanical, thermal, and photostabilities.

## Discussion

In contrast to many other color centers that are normally introduced in high-standard monocrystalline diamond, the purposely chosen SiV centers can be easily generated in scalable polycrystalline diamond microparticles at a much lower cost[64,65]. In fact, the Si substrate also serves as the Si doping source, and therefore we could "kill two birds with one stone". Particularly, these in-situ incorporated SiV centers have been shown to possess superior fluorescence properties compared to that generated by ion implantation[66]. In addition, the NIR emission (~737 nm) of SiV center would reduce the difficulty of distinguishing confidential from commonly encountered noise in visible range. Moreover, the highly emissive SiV centers would enable a wide

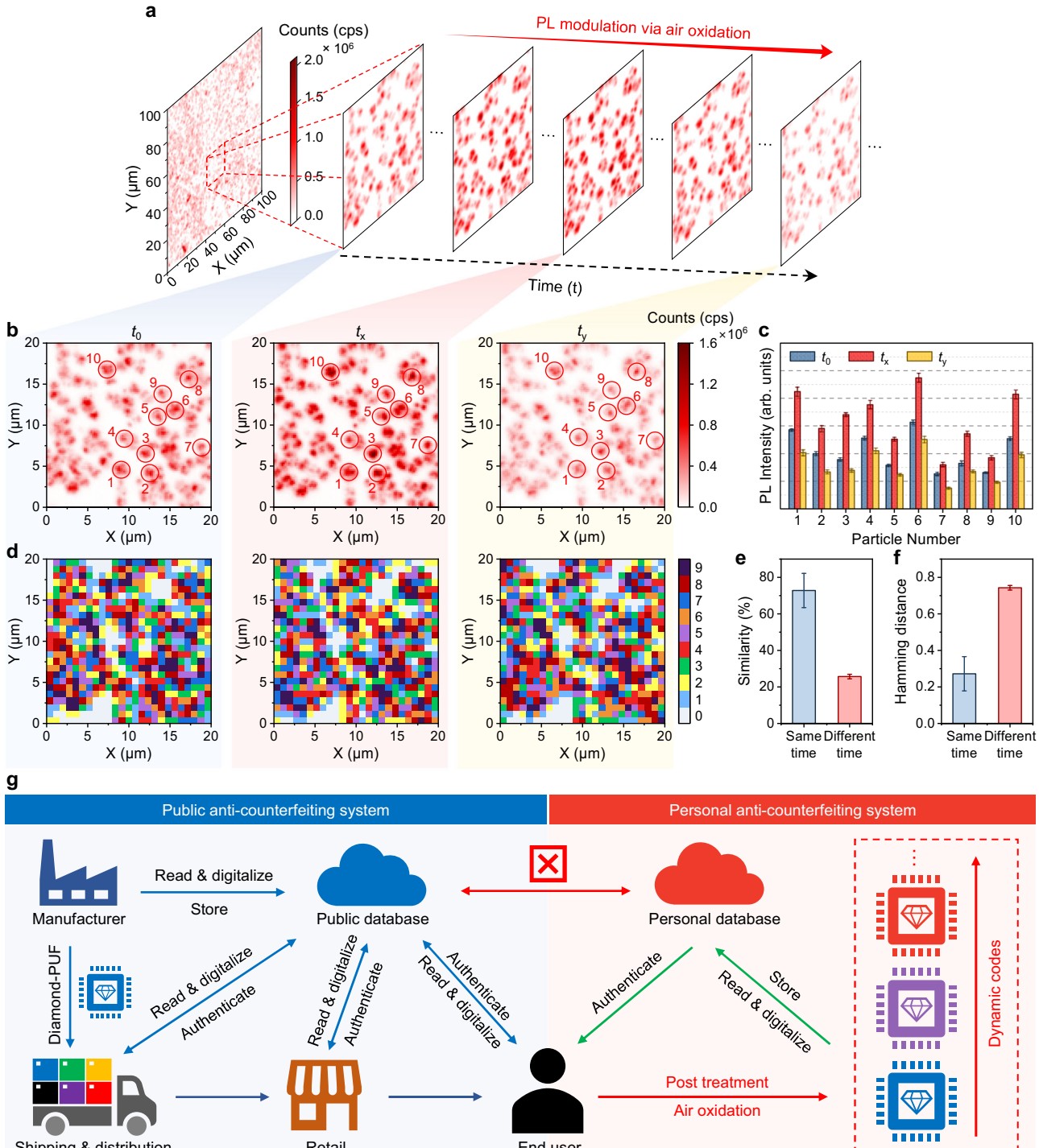

**Fig. 5 | Dynamic encoding based on PL intensity modulation via air oxidation.**
**a** A confocal fluorescence image of the diamond-PUF label with an area of 100 μm × 100 μm. **b** The confocal fluorescence images of the red dashed area of 20 μm × 20 μm in (**a**) at 3 different times ($t_0$, $t_x$, $t_y$). The color bar is kept the same for different images. **c** The PL intensity of 10 random diamond microparticles circled in (**b**) at three different time ($t_0$, $t_x$, $t_y$). **d** The digitization of decimal encoding of the area shown in (**b**) based on the PL intensity at each pixel, at a resolution of 25 pixels × 25 pixels. **e** The similarity index and **f** Hamming distance (normalized) of the decimal keys generated at the same and different times. **g** Demonstration of the utilization of the diamond-PUF labels in public and personal anti-counterfeiting systems. The error bars in **c**, **e** and **f** represent standard deviation from at least three independent measurements.

distribution of measured PL intensities (Fig. 4c), which would lead to a higher encoding capacity desired for anti-counterfeiting applications.

Scalable fabrication at low cost is critical in practical applications of PUF anti-counterfeiting labels. Accompanied with rapid development in recent years, the CVD approach has been recognized as the most convenient and cost-effective way for large-

scale commercialization of heterogeneously grown polycrystalline diamond on various substrates[49,65]. Notice that our adopted polycrystalline diamond microparticles are believed to have a much lower manufacturing cost than that of the single-crystal counterparts[65]. We have made a mild estimation of the total cost of fabricating the diamond PUFs, with a value of just ~0.0001 USD

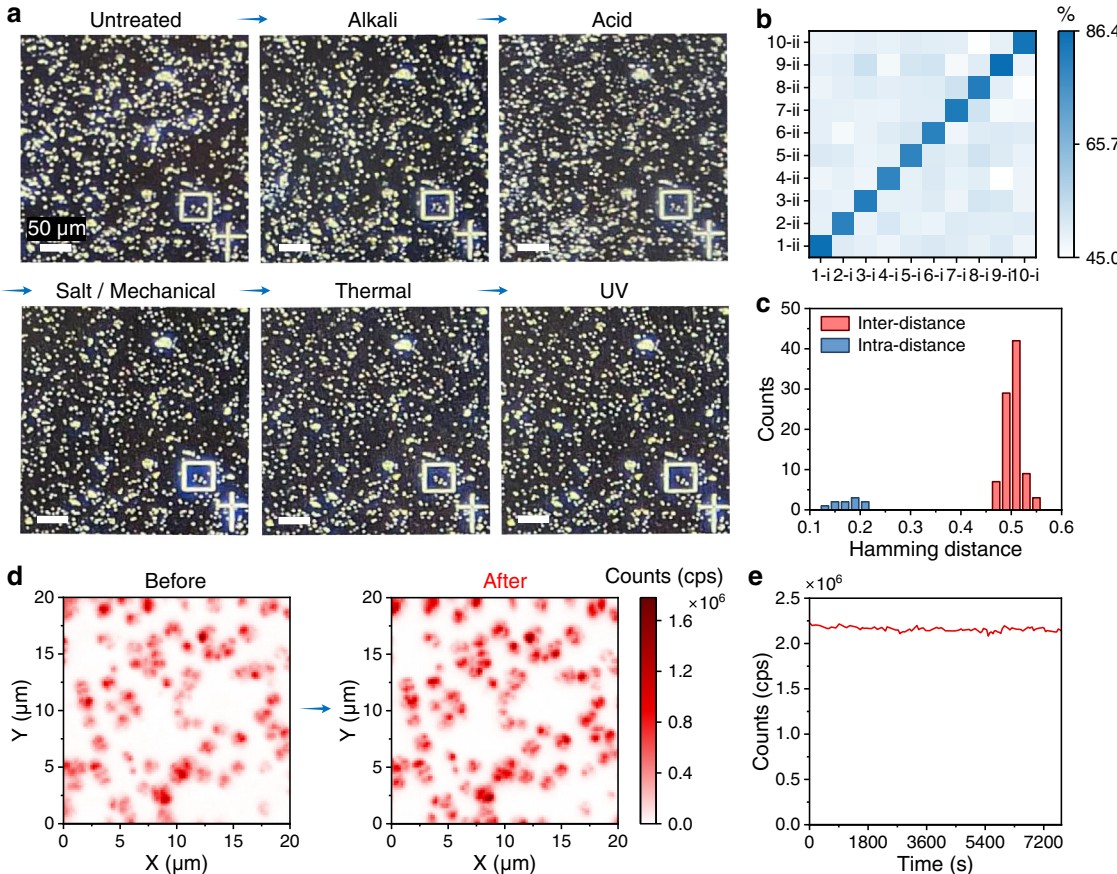

**Fig. 6 | The stability of the fabricated diamond-based PUF labels. a** The optical microscopy images of the same area on the diamond-PUF before and after a series of extreme treatments: (1) incubated in 1 M NaOH for 24 h; (2) incubated in 1 M HCl for 24 h; (3) milled with NaCl particles and incubated in supersaturated NaCl for 24 h, then washed with water under sonication (1 h); (4) heated at 400 °C for 24 h; and (5) exposed to ultraviolet (UV) light for 2 h. **b** The pairwise match of ten diamond-PUF labels before and after the treatments. The x-axis and the y-axis represent the first and the second measurement of the labels, respectively, and the color bar shows the similarity index. **c** The distribution of Hamming distances (normalized) of the keys generated from the ten diamond-PUF labels before and after the treatments. **d** The confocal fluorescence images of the same area on a diamond-PUF chip before and after a series of extreme treatments. **e** The photo-stability test (2 h) of the SiV centers in a diamond microparticle.

per PUF label (as shown in Supplementary Table 2 and Supplementary Note 3). In fact, the processing of the wafer-based diamond-PUF could be well suited by low-cost fabrication techniques (e.g., standardized laser cutting, lithography, etching, etc.) used in light-emitting diode (LED) industry, chip fabrication, and related production lines.

Moreover, our diamond PUFs anti-counterfeiting system could be upgraded with artificial intelligence (AI) technologies to improve authentication process[13,67,68]. For example, some AI algorithms could be applied for the rapid and accurate authentication of the diamond-PUFs by comparing the "fingerprint" of shape features and light scattering spectra of the diamond particles. Furthermore, actual perturbations (e.g., positioning angle, lighting conditions, magnification, and poor focus) must be considered in practical applications. Therefore, robust AI-assisted authentication is critical for practical usage.

In summary, we propose and demonstrate an advanced anti-counterfeiting material, i.e., diamond microparticles with SiV centers on Si wafer by CVD fabrication. Our findings offer the welcome prospect of the immediate development of a robust, dynamic, trustworthy, and intelligent anti-counterfeiting strategy. The PUF nature of the light scattering patterns, spectra, shape, and the SiV PL intensity of the diamond microparticles offer high-capacity static encoding. The properties of the SiV PL and the packed patterns of diamond microparticles are modulated upon air oxidation, offering the dynamic

security of the diamond PUFs. Additionally, the excellent stability of the diamond PUFs towards chemicals, mechanical forces, thermal heating, and UV exposure, indicates their great potential for use in practical applications.

## Methods
### Preparation of CVD seeds
0.1 g nanodiamonds (NDs) with a mean particle size of 50 nm (HPHT, PolyQolor, China) were mixed with 0.5 g sodium chloride (NaCl, 99.5%, Sigma-Aldrich) and heated at 500 °C for 1 h in air. The resultant sample was dispersed in deionized (DI) water and sonicated for 1 h, and the NDs were then purified with DI water three times by centrifugation. The purified NDs were re-dispersed in DI water and sonicated for 10 min to obtain a well-dispersed NDs suspension for the CVD diamond growth.

### Fabrication of diamond microparticles on the Si substrate by CVD
The Si substrate was firstly treated with hydrogen plasma by the microwave-plasma assisted chemical vapor deposition (MPCVD) system (Seki 6350) for 10 min. The prepared CVD seeds were then spin-coated on standard single-crystal Si (100) wafers (2 inches). The diamond microparticles were grown with a gas mixture of $H_2/CH_4$ (94/6) under fixed power (3400 W), pressure (85 torr), and temperature (920 °C) conditions for 80 min.

## FDTD simulations

The numerical simulations of scattering spectra of diamond microparticles were performed using a commercial FDTD software (Lumerical FDTD Simulation). In simulations, a total-field scattered-field source in wavelength range of 300–800 nm was launched into a box containing a diamond structure. The diamond structure and the surrounding space were divided into 2-nm meshes for calculating the scattering spectra.

## Air oxidation of the CVD-grown diamond microparticles

SiV PL modulation: the as-grown diamond microparticles were oxidized in air to modulate the PL intensity of SiV centers at 600 °C for 15 min, and the process was repeated 4 times. Diamond pattern transformation: the as-grown diamond microparticles were oxidized in air at 650 °C for 5 h, and the process was repeated 3 times.

## The stability of the diamond-PUF labels

Chemical stability: the diamond tag was incubated in 1 M NaOH, 1 M HCl, supersaturated NaCl solutions for 24 h, respectively, and then washed with clean water. Thermal stability: the diamond tag was heated in the air from room temperature to 400 °C and kept for 24 h. Mechanical stability: the diamond tag was mixed with salt particles, physically moved around for 30 min, and then washed with water under sonication (1 h). Photostability: the diamond tag was exposed to UV light (wavelength: 365 nm) for 2 h. After each treatment, the functionality of the diamond tag was characterized. Photostability of SiV centers: the PL intensity (counts: cps) of a diamond microparticle was collected for 2 h by exciting it with a 532 nm laser.

## Characterizations

The shape of the diamond microparticles was analyzed by scanning electron microscopy (SEM, Hitachi S-4800). The elemental composition of the grown diamonds was analyzed by electron energy loss spectroscopy (EELS) using a scanning transmission electron microscope (STEM, JEM-2100F). The crystallinity of the grown diamond was analyzed using X-ray diffraction (XRD, D8 Advance, Bruker). The Raman spectrum of the sample was measured by a Renishaw InVia Raman microscope. Single-particle dark-field scattering spectrum was recorded on an Olympus BX51 upright microscope that consists of a quartz-tungsten-halogen lamp (100 W), a monochromator (Acton, SpectraPro 2360i), and a charge-coupled device camera (Princeton Instruments, Pixis 400, cooled to −70 °C). PL characterizations of the sample were done using a home-built confocal microscope based on a nanopositioning stage (P-562.3CD, Physik Instrumente). In the excitation path, a continuous-wave 532 nm laser (polarization ratio > 100:1) is used to excite SiV centers with an oil immersion objective (NA 1.45 UPLXAPO100XO, Olympus). The fluorescence signal is collected in the detection path by single-mode fused fiber optic couplers (TW670R5F1, Thorlabs) with two single photon counting modules (SPCM-AQRH-16-FC, Excelitas Technologies) as detectors. A 647 nm long-wave pass filter was employed before detection to filter out the scattering signal.

## Readout and digitization of the diamond PUFs

Diamond PUFs were read by (1) a portable microscope with a smartphone attached, and (2) a home-built confocal microscope. To transduce the binary, quaternary, decimal and hexadecimal code matrixes from the optical and fluorescence images, MATLAB (R2021b) codes were designed for digital key extraction and performance analysis. The calculation process and equations for the uniformity, similarity index and Hamming distance of the PUF keys are shown in Supplementary Note 4.

## Data availability

The authors declare that the data supporting the findings of this study are available within the article and its Supplementary Information file.

All other data that support the findings of the study are available from the corresponding author upon request. Source data are provided with this paper.

## Code availability

MATLAB (R2021b) codes used for processing and digitizing the data are provided together with the source data in the Source Data file.

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

## Acknowledgements

Z.Q.C. acknowledges financial support from the HKSAR Research Grants Council (RGC) Early Career Scheme (ECS, No. 27202919); HKSAR Research Grants Council (RGC) Research Matching Grant Scheme (RMGS, No. 207300313); HKSAR Innovation and Technology Fund (ITF) through the Platform Projects of the Innovation and Technology Support Program (ITSP, No. ITS/293/19FP); HKU Seed Fund (Nos. 202011159019 and 202010160007); and the Guangdong Special Support Project (No. 2019BT02X030). L.S. acknowledges support from the Pearl River Talent Recruitment Program (No. 2019QN01C216) and the Shenzhen Science and Technology Program (No. JCYJ20210324140805014). Q.W. acknowledges financial support from the Guangdong Basic and Applied Basic Research Foundation (2019B1515120081, 2019B1515120091). Y.Z. (Ye Zhu) acknowledges financial support from the Hong Kong Polytechnic University (No. 1-W149). Y.Z. (Yan Zhou) acknowledges support from the Guangdong Special Support Project (2019BT02X030). D.Y.L. acknowledges the financial support from Innovation and Technology Commission of Hong Kong (ITF grant reference No. GHP/026/19GD).

## Author contributions

Z.Q.C. and T.T.Z. conceived the idea and wrote the manuscript. Z.Q.C., L.S. and Q.W. supervised the project. T.T.Z. and Z.Q.W. prepared the materials. J.W. carried out dark-field scattering spectrum measurements and FDTD Simulations. T.T.Z. and L.Z.W. performed confocal fluorescence measurements. T.T.Z. and M.G. performed SiV PL spectrum measurements. X.Y.G. and Y.Z. (Ye Zhu) carried out STEM and EELS measurements. Y.C.Y. and T.K.C.H. performed Raman measurements. Y.Z. (Yan Zhou), C.L., D.Y.L., K.H.L., and X.Q.W. discussed the results and commented on the manuscript.

## Competing interests

Z.Q.C, Q.W., T.T.Z. and Z.Q.W. are inventors on the Chinese Invention Patent Application no. 202210835536X filed on 15 Jul 2022. The remaining authors declare no other competing interests.
