## [Peer Review File · Nature Communications]

Multimodal dynamic and unclonable anti-counterfeiting using robust diamond microparticles on heterogeneous substrateEditorial Note: Parts of this Peer Review File have been redacted as indicated to maintain the confidentiality of unpublished data.

REVIEWER COMMENTS

Reviewer #1 (Remarks to the Author):

The authors have investigated the possible application of SiV centres for unclonable labels. While the study is interesting I feel the authors have over emphasised its utility. the biggest draw back in this process is the CVD growth of diamond which takes place at very high temperature (\Rightarrow 800C). This fact has also been not mentioned in the script.

Also, it is not clear why SiV was chosen rather than NV which is much easier to fabricate, although, one would need high temperature. Considering this is not that big a breakthrough I feel it is suitable for more specialised journal rather than NComms.

Reviewer #2 (Remarks to the Author):

The authors present a anti-counterfeiting strategy by using their previously developed high-quality CVD-grown diamond microparticles on heterogeneous substrates. It is a interesting and novel work. However, this manuscript need to solve the following concerned issues before it is considered for publication.

1. To achieve PUFs, the uncertainty of the process is only as a security entry condition, how many methods are there in Fig. 1a(2)? It cannot be outlined by etc., it is not scientific and rigorous. If it is related to the encryption capacity, how sensitive is each method?
- 2.The authors claim: “the developed diamond PUFs exhibit excellent performance in respects such as capacity, diversity, safety, manufacturability, robustness and compatibility (Fig. 1b)”, What is the basis? Is there a consensus among other scholars? Qualitative or quantitative data support is necessary.
- 3.Since the author claims their excellent performance in respects such as capacity, diversity, safety, manufacturability, robustness and compatibility, The authors should provide one-to-one results data to support their views.
- 4.For the application of anti-counterfeit labels, how do users authenticate? What is the ease of use?
- 5.What is the resolution of Fig. 3a(ii)? How many times can the camera of an ordinary smartphone be enlarged to take such pictures? Are this kind of smartphone ordinary for most users?
- 6.The author is focus on anti-counterfeit labels, why they show some figures for the means of chaos key generation, which is not the same as anti-counterfeit labels, right?
- 7.Line 213, if for information encryption, decryption is required, please provide a possible decryption scheme.
- 8.The authors should distinguish between the concepts of encryption and anti-counterfeiting. And the reviewer think the authors need to clarify the role of the mentioned encryption in the manuscript?

Reviewer #3 (Remarks to the Author):

This manuscript demonstrated multimodal and dynamic PUF labels by using SiV diamond microparticles on Si wafers by CVD fabrication. Authors achieved (1) high anti-counterfeiting capacity due to multimodal optical information like light scattering spectra, shape, and PL intensity; (2) dynamic anti-counterfeiting strategy to enhance the security of the diamond by air oxidation;(3) outstanding stability and durability. The work is significant original progress and offer a new material system for PUF fields.

However, the problems are also evident as follows. Authors conclude that the system exhibits excellent performance in respects such as capacity, diversity, safety, manufacturability, robustness and compatibility, is low cost and can be practically applied immediately as anti-counterfeiting labels in diverse fields. However, the fabrication technique by high-temperature CVD, limited substrate (Si), and dynamic strategy by high temperature air oxidation seriously limit its practical application in cryptography.

Overall, this paper can be considered for publication if the authors address the relevant critical problems to support the conclusions.

- 1) The patterning techniques are critical for optical PUF anticounterfeiting, which can enable PUF with patterns concluding goods information or anticounterfeiting information (that can be authenticated conveniently), and offer the locations for authentication of micro/nano PUF. The patterning strategies of MPCVD PUF should be offered and discussed.
- 2) The novelty in diamond microparticles themselves should be explained, such as the preparation or customization for anti-counterfeiting applications.
- 3) In Figure 1b, the authors highlight diamond PUFs exhibit excellent performance in respects such as capacity, diversity, safety, manufacturability, robustness and compatibility. However, what data does the Figure rely on? How to qualify these? In addition, the data in the figure of plasmonic NPs, organic dots, quantum dots are missing/unrecognizable.
- 4) The calculated the similarity index (Fig. 3e) and Hamming distance are used to quantitatively demonstrate the uniformity, uniqueness, and randomness of the diamond PUFs. But the calculation process or equations are missing, and the parts should be offered in detail in the Manuscript or Methods.
- 5) Cost on the fabrication and substrate is much higher than the requirement for large-scale commercialization. The fabrication of the diamond PUFs need a high temperature (500°C) and the microwave-plasma assisted chemical vapor deposition (MPCVD) system (Seki 6350) on standard single-crystal Si (100) wafers (2 inches) . The total cost of the diamond PUFs should be calculated and compared.
- 6) In terms of dynamic encoding modulation of the SiV diamond microparticles PUF via air oxidation, the stimulation of air oxidation is high temperature 600°C lasting 15 minutes. That is, the practical application of dynamic encoding is challenging, if not impossible. In addition, the time-dependent encoding is relying on the dynamic emission intensity/brightness of diamond microparticles, which is prone to be affected by the ambient light and exposure time during capturing PUF images. How to deal with this, and the author should explain it.
- 7) The authors say that the anti-counterfeiting system could be further combined with machine vision, artificial intelligence, and machine learning to identify and track the diamond-PUF labels efficiently and automatically. However, the relative emission intensity/brightness of diamond microparticles with time changed synchronously (Figure 4b-d); and one of the advantages of machine vision, artificial intelligence, and machine learning is to identify the specific pattern with different brightness, so how to take full use of the dynamic emission intensity/brightness is important.
- 8) The transformation of capture to binary encoding image is the basis for calculation. Since digital key extraction and performance analysis were conducted by MATLAB, the codes should be offered for the work to be reproduced.
- 9) The “multimodal” and “dynamic” are key information and contribution, which can be highlighted and added in the Title.

POINT-BY-POINT REPLY TO REVIEWER COMMENTS

We thank all the reviewers for their valuable comments, which we have used to improve our work. Please find below the point-by-point reply to the comments, with the reply in blue.

REVIEWER COMMENTS

Reviewer #1 (Remarks to the Author):

The authors have investigated the possible application of SiV centres for unclonable labels. While the study is interesting I feel the authors have over emphasised its utility. the biggest draw back in this process is the CVD growth of diamond which takes place at very high temperature (=> 800C). This fact has also been not mentioned in the script.

Also, it is not clear why SiV was chosen rather than NV which is much easier to fabricate, although, one would need high temperature. Considering this is not that big a breakthrough I feel it is suitable for more specialised journal rather than NComms.

Response:

We are grateful for the reviewer acknowledging that our study is interesting, and would like to take this opportunity to highlight the novelty of our work:

- (1) **multimodal and dynamic** diamond-PUF encoding capability adaptable to various anti-counterfeiting requirements, e.g., rapid authentication and high-level security;
- (2) **high encoding capacity** of the diamond-PUF anti-counterfeiting label;
- (3) **non-toxic and eco-friendly** diamond material suitable for practical anti-counterfeiting products requiring high safety, e.g., food and medicine.
- (4) **ultrahigh stability** of the diamond-PUF in extreme application scenarios, e.g., harsh chemical environments, high temperature, mechanical abrasion, and UV light irradiation;
- (5) **scalable fabrication** of the diamond-PUF at low-cost by a one-step commercial CVD system;
- (6) CVD fabrication process of the diamond-PUF has **excellent compatibility** with different product fabrication lines, e.g., microelectronics.

We have tried our best to address the two major concerns raised by the reviewer. The details are as follows.

(1) About the CVD growth of diamonds at high temperatures:

We feel sorry that the reviewer might misunderstand our described CVD growth process of diamond particles. We have strengthened the corresponding description and added the technical details in the Methods section of the revised manuscript, see page 8, line 335. Indeed, our adopted CVD system operated at a temperature of ~900 °C (a commonly used condition for diamond growth as summarized in Table R1); however, we do not see significant side effects/drawbacks of CVD growth for our proposed diamond-PUF product/labels. In general, all the possible methods for growing diamond materials need to work under highly strict conditions (high temperature and high pressure, see Table R1). Compared with the other two commonly known fabrication approaches, detonation and high-pressure high-temperature (HPHT) processes, the CVD process is believed to be the most convenient way to heterogeneously grow large-scale high-quality diamond particles with different color centers on various substrates at a relatively low cost^{R1-6}. Moreover, a relatively high-temperature growth condition is generally favored in the modern large-scale fabrication of high-quality semiconductor materials^{R7, 8}, because it normally ensures the great stability of materials. Thus, the CVD growth of diamonds at relatively high temperatures is one strategy for fabricating highly stable diamond-PUF devices. Of course, when needed, the low-temperature CVD growth of diamond is also achievable^{R9-12}.

Table R1. Key growth parameters (temperature, pressure, etc.) of the commonly used diamond synthesis methods

Method	Temperature	Pressure	Remarks	Ref
HPHT	1500-2200 °C	7-10 GPa	Mainly used for bulk or microdiamonds synthesis. Metal catalysts (Fe, Ni, or/and Co) are used.	R13
Detonation	1727-2727 °C	5~200 GPa	Mainly used for nanodiamonds (4-5 nm) synthesis. The resulting sample contains contaminants of non-diamond graphitic carbon (25-85%), metal and metal oxides (1-8%).	R14
CVD	300-1000 °C (Substrate)	1-200 Torr (Gas)	Do not require high pressure. Used to produce high-quality diamonds on large substrates with little contamination and good controllability of the growth process. The method has the flexibility for the choice of substrate, growth rate, and in-situ doping options.	R3, 5

The most important motivation of this work is the **non-deterministic** nature of CVD process, including its seeding, nucleation, and crystal growth procedures, satisfying the critical

requirement for fabricating PUF labels. Thus, by taking advantage of this intrinsic property of the CVD process, we directly used the non-deterministic grown diamond microparticles on heterogeneous substrates as PUF anti-counterfeiting labels. In addition, we want to emphasize the great advantage of the one-step **heterogeneous** CVD growth manner on the substrates, which greatly guarantees the strong adhesion of diamond particles to the silicon substrate, enabling the excellent stability of our diamond-PUF label as a whole device. But those direct methods by dip/spin-coating pre-synthesized particles on substrates as PUF labels (e.g., splitting the materials synthesis and PUF device fabrication into two separate steps) may have stability concerns. From a practical point of view, advanced instruments (e.g., CVD system) and harsh growth conditions (e.g., high temperature) are highly preferred for manufacturing diamond-based anti-counterfeiting labels as replicating them becomes much more challenging (almost being impossible) for general public^{R15}.

(2) About the utilization of SiV centers rather than NV centers:

We strongly believe that the SiV center is much easier to fabricate compared with NV center during CVD diamond fabrication. Actually, the formation of SiV centers in CVD-grown diamond is rather simple, due to the residual silicon sources that are often present in CVD chambers from silicon-containing substrates (or quartz bell jars), as shown by us and many other researchers in the field^{R6, 16, 17}. In contrast to many other color centers that are normally introduced in the expensively grown monocrystalline diamond, the bright SiV centers can also be prepared in **polycrystalline diamond particles at a much lower cost and a larger scale**^{R2-4}, **which is exactly what we have done in this work**. In our designed experiments, we believe that the fabrication of SiV centers is more proper and convenient (than NV centers), given that the chosen Si substrate is also the Si doping source, and therefore we could “kill two birds with one stone”. Moreover, these *in-situ* incorporated SiV centers during CVD growth have been shown to possess superior fluorescence properties compared to that generated with ion implantation^{R18}. In contrast, additional processes must be performed to introduce NV centers in our diamond sample, for example, (1) during the CVD process: the adding of nitrogen dopants like nitrogen gas; and/or (2) after the CVD process: (i) the incorporation of nitrogen impurities by doping or implantation, (ii) the creation of lattice vacancies by electron or laser irradiation, and (iii) a final annealing process (e.g., ≥ 700 °C in high-vacuum or inert gas for several hours)^{R19}. Thus, the facile creation of high-quality SiV centers in our diamond-PUFs product by the CVD process, in a cost-effective and scalable manner, is one of the key reasons that we adopt SiV centers to demonstrate the anti-counterfeiting applications.

Apart from the easy introduction of SiV centers to our diamond particles, we also prefer the spectroscopic features of SiV centers rather than NV ones. As described in the introduction of the manuscript, “*In particular, the SiV center exhibits a naked-eye-invisible near-infrared (NIR) emission at around 737 nm⁴⁰, reducing the difficulty of distinguishing confidential from disturbance information¹⁴. Thus, the color centers with NIR emissions, like SiV centers in diamond, have great potential to serve as optical PUF labels.*”, the NIR emission (~737 nm) of SiV center would reduce the difficulty of distinguishing confidential from disturbance information, which is one of the reasons why the color centers with NIR emissions, like SiV centers in diamond were chosen in our project, rather than the NV centers with a broadband emission (ZPL~638 nm for negatively charged NV, and a large portion falls in the visible region). Moreover, the high photon emission rate of SiV centers (room-temperature emission rates of various single color centers in diamond are summarized in Table R2) would enable a wide distribution of the SiV PL intensities (Fig. 4c), which would lead to a larger encoding capacity of the proposed PL-based anti-counterfeiting applications.

Table R2. Room-temperature single-photon emission rates of typical color centers in diamond

Color center	ZPL (nm)	Emission rates (counts/s)	Ref
SiV in CVD-grown nanodiamond	737	$\sim 6.2 \times 10^6$	R20
NV in nanodiamond	637	$\sim 1.6 \times 10^6$ (enhanced by optical microcavities)	R21
GeV in single-crystal bulk diamond	602	$\sim 1.7 \times 10^5$	R22
SnV in single-crystal bulk diamond	620	$\sim 1.4 \times 10^6$	R23
PbV in single-crystal bulk diamond	552	$\sim 1.0 \times 10^6$	R24
Ni-related in CVD-grown nanodiamond	768	$\sim 2.0 \times 10^5$	R25
Cr-related in single-crystal bulk diamond	749	$\sim 5.0 \times 10^5$	R26

To avoid any possible confusion regarding the fabrication of our proposed diamond-PUF product/labels, and to emphasize the reasons for selecting SiV centers for anti-counterfeiting in our work, we have added the above discussion in the revised manuscript, see page 7, line, 282.

Reviewer #2 (Remarks to the Author):

The authors present an anti-counterfeiting strategy by using their previously developed high-quality CVD-grown diamond microparticles on heterogeneous substrates. It is an interesting and novel work. However, this manuscript needs to solve the following concerned issues before it is considered for publication.

Response:

We gratefully thank the reviewer's time, helpful comments, and appreciation of the novelty of our work. We have thoroughly addressed the reviewer's concerns in our following point-by-point response.

Q1 To achieve PUFs, the uncertainty of the process is only as a security entry condition, how many methods are there in Fig. 1a(2)? It cannot be outlined by etc., it is not scientific and rigorous. If it is related to the encryption capacity, how sensitive is each method?

Response:

We thank the reviewer for pointing this out, and apologize for this unclear presentation in Fig. 1a(2). A PUF is a physical object with an inherent, unique, and fingerprint-like feature generated in a non-deterministic process. Here, for step (2) "Non-deterministic CVD", we wanted to emphasize that (i) the CVD growth of diamond particles is a **non-deterministic** process, satisfying the critical requirement for fabricating PUF labels; (ii) the color centers, such as SiV, NV, germanium vacancy (GeV), tin vacancy (SnV) centers, could be **stochastically** and *in-situ* introduced into the diamond lattice during the CVD process, which is one of the key points of CVD-grown diamond, without any extra defect-centers generation processes. As a result, a large **parameter space** offered by CVD process, for example, the concentration, amount, and type of color centers; size, shape, and crystallinity of the diamond particles can be freely modulated by tuning the CVD-growth parameters.

As for the maximum number of anti-counterfeiting methods based on color centers in diamonds, it is highly dependent on the type of color centers introduced by the CVD process (Fig. 1a(2)). Until now, there are ~500 different color centers that have been discovered in diamond^{R27}, and it has been demonstrated that a large portion (e.g., SiV^{R28}, NV^{R29}, GeV^{R22}, SnV^{R30}, Ni-related^{R31} and Cr-related^{R6} color centers) could be *in-situ* introduced during CVD process by utilizing suitable substrates or precursors. Although we just demonstrate the optical anti-

counterfeiting using the SiV centers in diamond particles, other color centers (e.g., NV, GeV, SnV) in diamonds can also be employed in principle. Each method may have its own encoding capacity, relating to the intrinsic optical properties (e.g., quantum efficiency, photon emission rate, lifetime) of the used color centers. Because the current proposed color center-related anti-counterfeiting strategy is based on the PL intensity of the color centers, different types of color centers may have their own level of PL intensity according to their emission rates (room-temperature emission rates of various single color centers in diamond are summarized in Table R2). For example, the higher the emission rates of the color centers (which will induce a higher P in C(3) of Fig. 1c), the larger the encoding capacity of the used color centers will be. The ultrahigh room-temperature single-photon emission rate of the SiV centers is one of the important reasons why we used them for high-level PL-related anti-counterfeiting.

To be more rigorous, we have deleted “(e.g., SiV, NV, GeV, SnV, ...)” in Fig. 1a(2) and added the corresponding description to the revised manuscript, see page 3, line 141.

Q2 The authors claim: “the developed diamond PUFs exhibit excellent performance in respects such as capacity, diversity, safety, manufacturability, robustness and compatibility (Fig. 1b)”, What is the basis? Is there a consensus among other scholars? Qualitative or quantitative data support is necessary.

Response:

We thank the reviewer for raising a point that allows us to better explain our work. We made such a conclusion based on comparing our results with the most representative optical PUFs shown in the reported works. Previously, some review papers have reviewed the performance of the common anti-counterfeiting materials^{R32-38}. However, not every reported PUF has all the required quantitative data in the aspects we compared. Here, to support our conclusion, we have summarized the quantitative, semi-quantitative, or qualitative data (see Table R3) of the performance of the most studied optical PUFs with respect to capacity, diversity, safety, manufacturability, robustness, and compatibility from the related reported works. (To be more accurate, we changed the “organic dots” into “polymer dots”, as we intended to describe the *photoluminescent materials in the form of polymer nanoparticles with a size of 1–100 nm*^{R38}, and we believe “polymer dots” is more accurate.)

Table R3. The performance of the most representative optical PUFs in respect of capacity, diversity, safety, manufacturability, robustness and compatibility.

PUF system	Capacity	Diversity	Safety	Manufacturability	Robustness	Compatibility	Ref
This work	$C = R^m$ (R: number of responses per pixel; m: number of pixels) $R = S + (n + 1) \times P$ (S: Scattering responses; P: PL responses; n: oxidation time)	 Physical patterns Scattering spectrum PL intensity Dynamic change of PL intensity 	Non-toxic, excellent biosafety and eco-friendly ^{R39}	One-step commercial integrated CVD system	Ultrahigh stability in different extreme application scenarios, including harsh chemical environments, high temperature, mechanical abrasion, and UV light irradiation.	Compatible with microelectronics ^{R3} ; The method has the flexibility for the choice of substrate, growth rate, and in-situ doping options ^{R5} .	-
Plasmonic NPs (e.g., gold NRs)	$C = R^m$	 Physical patterns Scattering color Polarization-dependent response 	NA (Toxicity of gold NRs is dependent on their size, aspect ratio, and surface characteristics)^{R40}	 Chemical synthesis process Drop-casting or dip-coating 	NA (Good stability against oxidation; Good photostability; Environmental sensitive)^{R36}	The NPs can be drop-cast onto a variety of substrates. (The physical properties of a substrate may influence the compatibility of the substrate with the NPs) (Gold is not CMOS-compatible)^{R36}	R41
Plasmonic NPs (e.g., gold NPs embedded with Raman probe molecules)	$C = R^m$	 Physical patterns Raman intensity Type of NPs 	NA (Toxicity of gold NPs is dependent on their size, shape, and surface characteristics)^{R42}	 Chemical synthesis process Drop-casting 	NA (Good photostability and long-term stability)^{R43}	The label can be fabricated on Scotch tape and transferred onto the surface of various products afterwards; Small and large aggregations formed may affect its compatibility. (Solvent based strategy, may not be compatible with microelectronics)^{R38}	R44

Polymer dots (e.g., π-conjugated polymer semiconductors)	$C = R^m$	1. Physical patterns 2. PL (color, lifetime) 3. Raman/Infrared spectrum	NA (High biosafety, and low toxicity, eco-friendly)^{R38}	1. Chemical synthesis process 2. Spin-coating or drop-casting	Good photostability; Stable in humidity/water and mechanical abrasion. (High stability toward environmental degradation)^{R38}	Good applicability to substrates of varying degrees of surface energy and flexibility. (Many may have low water solubility; Solvent based strategy, may not be compatible with microelectronics)^{R38}	R45
Quantum dots (II–VI semiconducting quantum dots, e.g., CdSe/CdS/CdZnS)	$C = R^m$	1. Physical patterns 2. PL (multicolor inks)	NA (The material usually contains toxic ions, e.g., Cd, and not eco-friendly)^{R32}	1. Chemical synthesis process 2. Inkjet printing	NA (Photo-blinking)^{R32} (Surface modification may result in excellent acid and base resistance, photostability, and thermal stability)^{R38}	NA (Adaptable to various substrates; Solvent based strategy, may not be compatible with microelectronics)^{R38}	R46
Photonic crystals (e.g., PSMA nanospheres)	$C = R^m$	Structural colors (Conflict between the “regular arrangement” of photonic crystals and the “randomness” of PUF)	NA	1. Chemical synthesis process 2. Patterning process (e.g., self-assembly, and electrostatic interaction)	Nonfading, good durability	NA (Complex fabrication process may affect its compatibility. Each anti-counterfeiting mark requires special design and complex fabrication process to allow a high-security level)^{R32}	R47
Perovskite NPs (e.g., CsPbBr₃ NPs)	$C = R^m$	1. Physical patterns 2. PL (multicolor inks, lifetime)	NA (The material usually contains toxic ions, e.g., Pb, and not eco-friendly)	1. Chemical synthesis process 2. Electrohydrodynamic printing	NA (Poor photostability, poor thermal stability, and poor humidity stability)^{R38}	NA (Adaptable to various substrates; Solvent based strategy, may not be compatible with microelectronics)^{R38}	R48

Upconversion NPs (e.g., Mn^{2+} co-doped in $NaLnF_4$, Ln =lanthanide)	$C = R^m$	1. Physical patterns 2. Emission profiles (multicolor, lifetime, different excitations)	NA (Some lanthanide dopant may be toxic, e.g., Tb, and not eco-friendly) ^{R32}	1. Chemical synthesis process 2. Stamping (used as inks)	Long-term stability at ambient conditions (High chemical and physical stability toward environmental degradation because of the advantages of ceramics) ^{R38} (Thermal quenching effect) ^{R32}	NA (Large-scale controlled synthesis is lacked) ^{R32} (Adaptable to various substrates; Solvent based strategy, may not be compatible with microelectronics) ^{R38}	R49
--	-----------	--	--	---	--	--	-----

Capacity: encoding capacity, the maximum number of unique PUFs that can be produced.

Diversity: the number of encoding methods that can be provided.

Safety: the degree of harm or danger the PUF may cause to the environment and human.

Manufacturability: the degree to which a PUF can be effectively manufactured given its design, cost, and distribution requirements.

Robustness: the ability to tolerate perturbations that might affect the PUF function.

Compatibility: the ability to work together in harmony because of well-matched characteristics.

NRs: nanorods

NPs: nanoparticles

NA: not available

PSMA: polystyrene-maleic acid copolymer

To summarize, our diamond-based PUF shows excellent performance in respect of capacity, diversity, safety, manufacturability, robustness, and compatibility. Other optical PUF systems may suffer from drawbacks such as safety concerns, poor stability, complicated chemical synthesis processes, poor compatibility with microelectronics, etc. In our original manuscript, we plotted Fig. 1b under a unified standard by using “High”, “Medium”, and “Low” to semi-qualitatively compare their performance in every aspect. We now feel this is more or less a bit biased due to the un-unified and uncompleted data sets available, therefore, we decide to remove Fig. 1b, and directly show the summarized Table R3 to the audiences, leaving them to be the judge. In replacement of the original Fig. 1b, we have highlighted the novelty of our work as Fig. R1.

Fig. R1. The favorable features of the developed diamond-PUFs.

We have added Table R3 as Supplementary Table 1 to the revised Supplementary Information, see page S-2, replaced Fig. 1b with Fig. R1, and revised the corresponding description in the revised manuscript, see page 2, line 97.

Q3 Since the author claims their excellent performance in respects such as capacity, diversity, safety, manufacturability, robustness and compatibility, The authors should provide one-to-one results data to support their views.

Response:

We thank the reviewer for pointing this out. Please see our response to Q2. We have provided a summarized table (Table R3) showing one-to-one quantitative, semi-quantitative, or qualitative data to support our point. From the table, we can clearly see that our diamond-based PUF shows good performance in respect of capacity, diversity, safety, manufacturability, robustness, and compatibility.

Q4 For the application of anti-counterfeit labels, how do users authenticate? What is the ease of use?

Response:

We thank the reviewer for pointing this out. The authentication process and ease of use of a PUF label are essential to its anti-counterfeiting applications. As shown in Fig. 5g of the manuscript, we presented the workflow of the anti-counterfeiting applications using our diamond-PUF label, but it might not be clear enough. Here, to better demonstrate the anti-counterfeiting application, we further provide a detailed explanation of the authentication process and ease of use of our diamond-PUF anti-counterfeiting label as follows.

Fig. R2. Schematic of authentication protocol based on diamond-PUFs, showing registration and authentication processes.

As shown in Fig. R2. The execution of authentication can be divided into two steps, including the registration and validation processes.

- (1) In the registration stage, input challenges (C_1) are projected onto the diamond-PUF to generate the associated response (R_1) , which is further transformed into the key (K_1) .

Then, the challenge-response pairs (CRPs) composed of the registered information of C_1 as well as K_1 are stored in the cloud database.

- (2) In the validation stage, the authenticator randomly selects C_1 from the cloud database to challenge the candidate PUF, and then obtains a check code K_1' . Hereafter, the check code is uploaded to the data cloud to compare the similarity index (I) or Hamming distance between K_1 and K_1' . If $I(K_1, K_1')$ is greater than a preset threshold, the verification is true. Otherwise, the verification result is false.

For ease of use, the smartphone-based authentication setup shown in Fig. 3a is expected to have the capability of being detected and read out quickly, cheaply, and conveniently. Due to their high refractive index, the diamond particles on the diamond-PUFs can be easily photographed by any smartphone equipped with a magnifying lens (200~300×). Therefore, the user can readily take and upload the photo to the cloud database for the further authentication process.

To better explain the anti-counterfeiting application, we have added the above discussion about the authentication process and ease of use of our diamond-PUF tag in the revised Supplementary Information as Supplementary Fig. 4 and Supplementary Note 1, see page S-5.

Q5 What is the resolution of Fig. 3a(ii)? How many times can the camera of an ordinary smartphone be enlarged to take such pictures? Are this kind of smartphone ordinary for most users?

Response:

We thank the reviewer for asking these detailed questions. As indicated in Fig. 3a(ii), the actual size of the red dashed square is $200\ \mu\text{m} \times 200\ \mu\text{m}$, which is calibrated from the near cross marker ($50\ \mu\text{m} \times 60\ \mu\text{m}$, fabricated by focused-ion-beam milling). And we used a very portable microscope (with 300× magnification) attached to the camera of an ordinary smartphone to take such pictures. The portable microscope is used for enlargement, and the smartphone is just used to take pictures. This kind of portable microscope is widely available in any stationery store at a relatively low price (we brought it for ~15 USD)^{R50}, which can be in principle attached to any kind of commonly seen smartphone. Thus, we believe this portable smartphone-based approach is suitable for most users.

Q6 The author is focus on anti-counterfeit labels, why they show some figures for the means of chaos key generation, which is not the same as anti-counterfeit labels, right?

Response:

We thank the reviewer for raising this technical point. When designing a PUF anti-counterfeiting system, the digitization (key generation from the readout) of the label is also essential, because the information content of the system is eventually stored in the unique codes generated in such a process. And one of the key points that we want to introduce in our manuscript is the **multimodal PUF** anti-counterfeiting functions of our diamond-PUF label, ranging from static to dynamic, low-level to high-level encoding, which means that we could have different ways to authenticate our diamond-PUF for different application scenarios. Each level of encoding has its own readout method, for example, physical distribution, scattering spectrum, and PL intensity of the diamond particles, the digitization (means of chaos key generation) process will then be different from each other, depending on the readout method. The low-level encoding based on physical patterns of diamond particles can be readout quickly, cheaply, and conveniently through mobile devices, which is applicable to applications requiring rapid authentication. At the same time, the high-level encoding (based on the scattering spectrum or PL intensity of diamond particles) requires precision instruments to read, but provides a more secure method, which is applicable to applications requiring high-level security. Therefore, to elaborate and evaluate our claimed multimodal function, we presented the results of chaos key generation from different means.

Q7 Line 213, if for information encryption, decryption is required, please provide a possible decryption scheme.

Response:

We thank the reviewer for asking this technical point. Here, we would like to propose a possible information encryption and decryption scheme using our diamond-PUF label.

Secure communication between two parties is important for everyday information security. Here, our diamond-PUF label has the application potential in secure information encryption. As shown in Fig. R3, a possible information encryption and decryption scheme using our diamond-PUF label is proposed. Firstly, two unique diamond-PUFs (A and B) are used to generate random and secret keys (K_A and K_B), which belong to the encryption and decryption

side, respectively. The two keys are then mixed and stored in a public database ($K_A \oplus K_B$), shown in Fig. R3a. During the information encryption stage, the diamond-PUF key K_A is used to encrypt the message (M) to generate the ciphertext ($M \oplus K_A$), which is then transmitted to the decryption side *via* a public channel. As for the information decryption, diamond-PUF key K_B is then used to decrypt the ciphertext by mixing the public database with the key K_B , i.e., $K_B \oplus (K_A \oplus K_B) \oplus (M \oplus K_A) = M$. Due to the unique and irreproducible nature of the diamond-PUF labels, there is no need for secret key sharing and storage during the encryption and decryption processes, ensuring efficient and secure information communication.

Fig. R3. Schematic of a possible information encryption and decryption scheme using the diamond-PUFs.

To better explain the information encryption application, we have added the above discussion about the possible information encryption and decryption scheme using our diamond-PUF

label in the revised Supplementary Information as Supplementary Fig. 5 and Supplementary Note 2, see page S-6.

Q8 The authors should distinguish between the concepts of encryption and anti-counterfeiting. And the reviewer think the authors need to clarify the role of the mentioned encryption in the manuscript?

Response:

We thank the reviewer for pointing out this unclear description. We apologize for our nonspecific usage of “encryption” which has been mixed with “anti-counterfeiting” in the manuscript. Our intention of mentioning encryption is to remind other possible PUF-related applications using our diamond-PUF tags besides anti-counterfeiting, like information encryption. To clarify this point, we have revised our “encryption” related description in the revised manuscript, as follows,

-Page 1, line, 28, “encryption” was changed to “anti-counterfeiting”:

“Developing advanced anti-counterfeiting materials ...”

-Page 1, line, 35, “encryption” was changed to “encoding”:

“... high-capacity optical encoding.

-Page 1, line, 35, “encryption” was changed to “encoding”:

“Moreover, time-dependent encryption encoding ...”

-Page 2, line, 71, “encryption” was changed to “encoding”:

“... next-generation anti-counterfeiting technologies ...”

-Page 3, line, 102, “encryption” was changed to “encoding”:

“... multidimensional optical encoding with a large capacity (Fig. 1c).”

-Page 3, line, 107, “encryption” was changed to “encoding”:

“The multimodal, dynamic encoding capability ...”

-Page 5, line, 200, “information encryption” was changed to “anti-counterfeiting”:

“... high-level anti-counterfeiting method.”

-Page 5, line, 202, “encryption” was changed to “readout”:

“... light scattering signals for readout.”

-Page 5, line, 206, “encryption” was changed to “encoding”:

“... achieve high-level encoding using the PL signals from the color centers ...”

-Page 5, line, 233, “encryption” was changed to “encoding”:

“... advanced time-dependent encoding method.”

-Page 6, line, 252, “encrypted information” was changed to “encoded keys”, and line 301, “information encryption” was changed to “anti-counterfeiting”:

“... allows the encoded keys of the same PUF label to be personally changed when the original label faces malicious attacks and the risk of being duplicated, a desirable feature in the design and fabrication of anti-counterfeiting material with high-level security.”

-Page 7, line, 310, “information encryption” was changed to “anti-counterfeiting”:

“... an advanced anti-counterfeiting material...”

-Page 7, line, 315, “encryption” was changed to “encoding”:

“... high-capacity static encoding.”

-Page 13, line, 588, “encryption” was changed to “anti-counterfeiting”:

“... dynamic features for advanced anti-counterfeiting.”

Reviewer #3 (Remarks to the Author):

This manuscript demonstrated multimodal and dynamic PUF labels by using SiV diamond microparticles on Si wafers by CVD fabrication. Authors achieved (1) high anti-counterfeiting capacity due to multimodal optical information like light scattering spectra, shape, and PL intensity; (2) dynamic anti-counterfeiting strategy to enhance the security of the diamond by air oxidation;(3) outstanding stability and durability. The work is significant original progress and offer a new material system for PUF fields.

However, the problems are also evident as follows. Authors conclude that the system exhibits excellent performance in respects such as capacity, diversity, safety, manufacturability, robustness and compatibility, is low cost and can be practically applied immediately as anti-counterfeiting labels in diverse fields. However, the fabrication technique by high-temperature CVD, limited substrate (Si), and dynamic strategy by high temperature air oxidation seriously limit its practical application in cryptography.

Overall, this paper can be considered for publication if the authors address the relevant critical problems to support the conclusions.

Response:

We are grateful for the reviewer acknowledging our work is significant original progress, and we have carefully addressed her/his concerns in our following point-by-point response.

Q1 The patterning techniques are critical for optical PUF anticounterfeiting, which can enable PUF with patterns concluding goods information or anticounterfeiting information (that can be authenticated conveniently), and offer the locations for authentication of micro/nano PUF. The patterning strategies of MPCVD PUF should be offered and discussed.

[Response redacted]

[Response redacted]

Q2 The novelty in diamond microparticles themselves should be explained, such as the preparation or customization for anti-counterfeiting applications.

Response:

We thank the reviewer for this valuable comment. Our large-scale fabrication of high-quality diamond microparticles with SiV centers on heterogeneous substrates offers excellent opportunities for anti-counterfeiting applications: (1) the flexible variability of CVD growth parameters (e.g., gas composition and flow rate, microwave power, pressure, growth time and temperature, diamond seeds, and substrates) and the wafer-scale production (substrate stage

with a diameter of 10 nm or even higher) ability make the mass commercial customization of the diamond-PUF labels possible; (2) the micro-size and high refractive index of the diamond particles could enable the convenient readout and authentication processes *via* a simple portable smartphone-based equipment (Fig. 3a); (3) the abundant amount of SiV centers introduced into the diamond microparticles together with the distinctive geometry features of the particles can be utilized for high-level anti-counterfeiting by analyzing their PL and light scattering signals; (4) the intrinsic excellent robustness (outstanding chemical inertness, excellent mechanical strength, and high-temperature stability) of the diamond microparticles is another crucial advantage for practical anti-counterfeiting applications; (5) the one-step heterogeneous CVD growth manner of the diamond particles on the substrates (rather than direct dip/spin-coating pre-synthesized particles on substrates as PUF labels) greatly guarantees the strong adhesion of diamond particles to the silicon substrate, which further enables the excellent stability of our diamond-PUF label as a whole device.

To clarify this point, we have added the corresponding discussion to the revised manuscript, see page 3, line 124.

Q3 In Figure 1b, the authors highlight diamond PUFs exhibit excellent performance in respects such as capacity, diversity, safety, manufacturability, robustness and compatibility. However, what data does the Figure rely on? How to qualify these? In addition, the data in the figure of plasmonic NPs, organic dots, quantum dots are missing/unrecognizable.

Response:

We thank the reviewer for pointing this out. We made such a conclusion based on comparing our results with other most studied optical PUFs shown in the reported works. Please kindly see our response to Q2 of Reviewer #2. We have provided a summarized table (Table R3) showing one-to-one qualitative, semi-quantitative, or quantitative data to support our point. To be unbiased (due to the lack of unified and completed data), we decided to remove Fig. 1b and directly show the summarized Table R3 to the audiences, leaving them to be the judge. In replacement of the original Fig. 1b, we have highlighted the novelty of our work as Fig. R1.

Q4 The calculated the similarity index (Fig. 3e) and Hamming distance are used to quantitatively demonstrate the uniformity, uniqueness, and randomness of the diamond PUFs.

But the calculation process or equations are missing, and the parts should be offered in detail in the Manuscript or Methods.

Response:

We thank the reviewer for pointing this out. The calculation process and equations for the corresponding parameters are as follows,

(1) *Uniformity*. The uniformity metric can be calculated using the following equation:

$$\text{Uniformity} = \frac{1}{n} \sum_{i=1}^n R_i$$

where, R_i is the i th bit response from the n -bit key.

(2) *Similarity index*. The similarity index (I) is used to detect the degree of similarity between different PUFs, which can be calculated by the following equation:

$$I = \frac{A}{B} \times 100\%$$

where, A is the same number of pixels between two PUFs, B is the total number of pixels in PUF.

(2) *Hamming distance*. The Hamming distance between two keys is the minimum number of substitutions required to change one key into the other. The Hamming inter-distance is used to quantify the uniqueness of a PUF, which is the ability to distinguish a PUF from others. And the Hamming intra-distance is used to quantify the reliability of the same PUF label, checking if it has the ability to generate consistent keys under multiple measurements. The uniqueness and reliability are evaluated using the Hamming distance as below:

$$\text{Uniqueness} = \frac{2}{N(N-1)} \sum_{i=1}^{N-1} \sum_{j=i+1}^N \frac{HD(R_i, R_j)}{n}$$

where R_i and R_j are n -bit keys of the i th and j th PUF, and N is the total number of PUFs.

$$\text{Reliability} = \frac{1}{N} \sum_{i=1}^N \frac{HD(R_0, R_i)}{n}$$

where R_0 is the original n -bit key of the PUF, and R_i is the n -bit key generated by the same PUF from i th measurement among total N times of measurements.

We have added the detailed calculation process and equations for the uniformity, similarity index, and Hamming distance as Supplementary Note 4 in the revised Supplementary Information, see page S-11.

Q5 Cost on the fabrication and substrate is much higher than the requirement for large-scale commercialization. The fabrication of the diamond PUFs need a high temperature (500°C) and the microwave-plasma assisted chemical vapor deposition (MPCVD) system (Seki 6350) on standard single-crystal Si (100) wafers (2 inches) . The total cost of the diamond PUFs should be calculated and compared.

Response:

We thank the reviewer for this very constructive comment. In general, lab-grown diamonds have existed for more than 60 years, and the synthetic technologies have been well-developed. The commercialization of CVD technology was realized in the early 2000s, and it has made crucial advances in recent years, allowing companies to grow higher-quality diamonds more rapidly and more cheaply. For example, the cost of producing a 1-carat G-color VS-polished CVD-grown single-crystal diamond (Fig. R5a) dropped significantly from **4000 USD** in 2008 to **300-500 USD** in 2018, with further reduction expected, according to the Global Diamond Report by the Antwerp World Diamond Centre^{R58}. At the same time, the global number of CVD systems (Fig. R5b) was **~5300** in 2020, and it is expected to significantly increase to **~31900** by 2025, which is mainly due to the manufacturing and assembly technology of CVD equipment has been mastered by more and more enterprises (such as companies in China, India, and other countries), according to a report by the Sinolink Securities^{R59}. Therefore, the cost of CVD equipment and fabrication is expected to continuously decrease along with significant technological advancement.

Fig. R5. a Production cost of 1-carat G VS polished CVD-grown single-crystal diamond (data from Antwerp World Diamond Centre^{R58}). **b** The estimated global number of CVD system (data from Sinolink Securities^{R59}).

In this study, we demonstrated the PUF anti-counterfeiting application by using the heterogeneously grown **polycrystalline** diamond microparticles, which are believed to have a much lower manufacturing cost than that of the single crystal counterparts^{R3, 5}. Our diamond-PUF samples were prepared by a Seki 6350 MPCVD system (purchased in 2018), which is indeed expensive (432000 USD). Note that the price of a MPCVD system has been dramatically reduced (~100000 USD) in the market for the low-cost and large-scale commercial fabrication of diamonds (see our uploaded file *Supporting Information_MPCVD*, showing the quotation and technical description for a standard MPCVD system similar to Seki 6350, which could fully meet all requirement for the fabrication of diamond-PUF labels). Most importantly, the total cost of fabricating the diamond-PUFs could be significantly and continuously reduced when mass production begins, because the MPCVD system is a one-time investment. In addition, we want to emphasize that the repeatability, yield rate, and production efficiency of the MPCVD technology (industrial level) are much higher than those laboratory level methods, which is greatly suitable for massive production. Nevertheless, it is actually beneficial to have complicated instruments (e.g., MPCVD system) and harsh growth conditions (e.g., high temperature) for manufacturing diamond-PUFs as replicating them becomes much more challenging (almost impossible) for the general public^{R15}.

Here, to provide a general estimation of the total cost of fabricating the diamond-PUFs, we have calculated the cost of fabricating one piece 2-inch sized diamond-PUF in almost all aspects based on our experiment and local situation, as shown in the following Table R4.

Table R4. The mildest estimation of detailed cost for fabricating one piece 2-inch sized diamond-PUF.

Item	Unit price	Consumption	Cost (USD)	Remarks
Electricity	0.11 USD/kWh	3.6 kWh	0.396	The average power of our MPCVD equipment during the fabrication process is 2.4 kW with a running time of 1.5 hours, and the local electricity price is 0.11 USD/kWh.
Gas (Hydrogen)	0.0017 USD/L	8.46 L	0.014	The average price of bottled hydrogen is 8.86 USD/bottle (13.5 MPa, 40 L), equal to 0.0017 USD/L (atm). During diamond growth, the hydrogen flow rate is ~0.094 L/min with a running time of 90 minutes.

Gas (Methane)	0.033 USD/L	0.48 L	0.016	The average price of bottled methane is 177 USD/bottle (13.5 MPa, 40 L), equal to 0.033 USD/L (atm). During diamond growth, the methane flow rate is ~0.006 L/min with a running time of 80 minutes.
Substrate	0.144 USD/piece	1 piece	0.144	The solar cell grade monocrystalline silicon wafer with a diameter of 2 inches and a thickness of 0.18 mm is used as the substrate for diamond growth in this study, and it can be purchased at a price of ~0.144 USD/piece from the market.
Diamond seeds	1.31 USD/g	0.0005 g	0.001	Diamond nanoparticles with a particle size of ~50 nm can be purchased at ~1.31 USD/g from the market. After the salt-assisted air-oxidation treatment, the diamond seeds are dispersed into deionized water with a concentration of ~0.1 wt%, and an average of 0.5 g of the seeds solution is spin-coated onto a 2-inch silicon wafer.
Equipment (MPCVD system)	0.846 USD/hour	1.5 hours	1.269	We used the current market price ~100000 USD for calculation. The MPCVD system will depreciate over 15 years with a utilization rate of ~90%. Then, the equipment purchase (one-time investment) can be converted into the running/maintaining cost ~0.846 USD/hour.
House rent	0.091 USD/hour	1.5 hours	0.137	The local house rent for placing the MPCVD system is ~720 USD/year. Considering the utilization rate (~90%) of the MPCVD system, the unit cost of house rent can be converted into ~0.091 USD/hour.
Labor	0.3 USD/hour	1.5 hours	0.450	The local labor cost is ~3 USD/hour. Considering the MPCVD system is highly integrated and does not require complex operations, one person can operate 10 or more equipment at the same time, so the unit labor cost can be converted into ~0.3 USD/hour. In the future, the labor cost will be lower or reduced to zero, because the final production line will develop into full-machine automatic production.
Total			2.427	This cost could be further reduced along with the significant technological advancement over the years.

The above calculated total cost is the unit price for fabricating one piece of 2-inch sized diamond-PUF label. Due to the high encoding capacity of the diamond-PUFs, a tiny working

area can meet the purpose of anti-counterfeiting. In our study, the largest area used for encoding is $200\ \mu\text{m} \times 200\ \mu\text{m}$ (Fig. 3a). Thus, a 2-inch wafer can be cut into a maximum of $\sim 4.9 \times 10^4$ pieces of working labels, and the cost of an individual working label will be significantly low (~ 0.0001 USD). And it will be further reduced if a larger-sized substrate is used for the diamond-PUF fabrication. In addition, we might need to consider the extra cost for processing (i.e., cutting the 2-inch wafer into many small pieces for our purpose) the Si wafer (~ 10 USD/piece), but this is negligible when considering the large number (on the order of $\sim 10^4$ pieces) of generated anti-counterfeiting labels from such a 2-inch wafer. In fact, the processing of the wafer-based devices could be well suited by low-cost fabrication techniques (e.g., standardized laser cutting, lithography, etching, etc.)^{R60-62} used in the LED industry, chip fabrication, and related production lines. Therefore, it can be predicted that the above-estimated cost of the diamond-PUFs (~ 0.0001 USD per label) would be well-accepted by high-end products with high anti-counterfeiting requirements, such as electronic components, medicine packaging, vehicles, luxury goods, etc.

In addition to the currently adopted Si wafer, it is possible to use many other substrates as well. It has been reported that diamonds can be CVD grown on other substrates, such as SiO_2 ^{R63, 64}, Al_2O_3 ^{R65}, GaN ^{R66, 67}, or even flexible substrates by later transformation^{R68, 69}. Moreover, many studies have successfully demonstrated the CVD fabrication of diamonds at relatively low temperatures^{R9-12}. As shown in Fig. R5 (our unpublished preliminary results, further quality improvements are needed), we have also demonstrated the CVD growth of diamond micro/nanoparticles on SiO_2 and sapphire (low growth temperature: $<475^\circ\text{C}$) substrates.

Fig. R6. SEM images of the CVD grown diamond micro/nanoparticles on **a** SiO_2 and **b** sapphire (low growth temperature: $<475^\circ\text{C}$) substrates. **Unpublished preliminary results.**

To clarify the above points, we have added the above discussion in the revised manuscript, see page 7, line 292, and added Table R4 as Supplementary Table 2 in the revised Supplementary Information, see page S-10.

Q6 In terms of dynamic encoding modulation of the SiV diamond microparticles PUF via air oxidation, the stimulation of air oxidation is high temperature 600°C lasting 15 minutes. That is, the practical application of dynamic encoding is challenging, if not impossible. In addition, the time-dependent encoding is relying on the dynamic emission intensity/brightness of diamond microparticles, which is prone to be affected by the ambient light and exposure time during capturing PUF images. How to deal with this, and the author should explain it.

Response:

We thank the reviewer's comment. For practical applications, we believe a good "threshold" for activating the dynamic encoding is very important for the stable function of the normal anti-counterfeiting ability, and dynamic coding is performed on demand when necessary. Fortunately, the proposed high-temperature air oxidation treatment is such good "threshold" by protecting the diamond-PUF from involuntary transformation (i.e., instability), for example, the diamond-PUF label can function stably in most practical application scenarios (Fig. 6), and it only changes into another new PUF key upon intentional treatment when the original label faces malicious attacks and the risk of being duplicated. At the same time, the air oxidation treatment (600°C, 15 minutes) could be easily achieved by any kind of CVD system, furnace, or heating plate, which is not a challenging requirement. One possible approach for the practical application of dynamic encoding is using the detachable strategy, e.g., detaching the diamond-PUF label for dynamic encoding modulation by air oxidation, and reattaching the label on the product after modulation.

As for the time-dependent encoding process, indeed, it depends on the emission intensity/brightness of the SiV centers in diamond microparticles. However, it was the difference between individual particles that we considered during each encoding process (Fig. 4c); for example, in the quaternary encoding: the intensities ranked between 0~25%, 25~50%, 50~75%, and 75~100% would be encoded as 0, 1, 2, and 3, respectively. And such intensity difference is intrinsically dependent on the SiV concentrations in diamond microparticles, which is determined by the CVD process. The influence of the ambient light and exposure time on the SiV emission is a kind of bulk manner (i.e., the intensity of each particle will increase

or decrease to almost the same extent); thus, it will not greatly affect each self-compared encoding process. Here, we have performed additional experiments (Fig. R7) to prove this point by reading out the same label under different conditions (with or without ambient light, exposure time, and laser power). From the results, we can know that the measurement conditions while capturing the PL images may not affect the final keys generated by the same diamond-PUF label because their similarity indexes are within an acceptable value ($>80\%$). Of course, to ensure the accuracy of authentication, we may try to keep the measurement condition of the readout process unchanged.

Fig. R7. The PL images of the same diamond-PUF label in a standard registration condition and different measurement conditions (b with and without ambient light, c different exposure time, and d different laser power) for validation. e The similarity index (%) of the quaternary keys generated by the same diamond-PUF label at different measurement conditions. (Note: the condition of the registration process: no ambient light, 5 ms exposure time, 0.5 mW laser power. For the validation process, the conditions other than those tested remain unchanged, e.g., the exposure time and laser power remain unchanged (5 ms, 0.5 mW) when testing the effect of ambient light.)

To clarify these points, we have added the above discussion in the revised manuscript, see page 5, line 224, and added Fig. R7 as Supplementary Fig. 8 in the revised Supplementary Information, see page S-8.

Q7 The authors say that the anti-counterfeiting system could be further combined with machine vision, artificial intelligence, and machine learning to identify and track the diamond-PUF labels efficiently and automatically. However, the relative emission intensity/brightness of diamond microparticles with time changed synchronously (Figure 4b-d); and one of the advantages of machine vision, artificial intelligence, and machine learning is to identify the specific pattern with different brightness, so how to take full use of the dynamic emission intensity/brightness is important.

Response:

We thank the reviewer for pointing this out, and believe there might be some misunderstanding of AI-related discussion. Thus, we have revised the corresponding part so as to make it clear to the audience. We definitely agree with the reviewer that the strength of machine learning is to identify the specific pattern with different brightness, and our intention was to say that those AI techniques could be combined with any **fixed** diamond-PUF anti-counterfeiting labels to further improve the overall performance. For example, some AI algorithms could be developed for the rapid and accurate authentication of the diamond-PUF label when using the anti-counterfeiting based on the shape feature and scattering spectrum of the diamond particles (e.g., one can employ an “internal reference” of a diamond particle with fixed size and shape for the signal normalization). Furthermore, some noise factors (e.g., positioning angle, lighting conditions, magnification, and poor focus) should be considered in the real scenario; thus, robust AI-assisted authentication is desired for practical usage.

As for the mentioned dynamic features in emission intensity/brightness, we actually applied air oxidation treatment (Fig. 5a-d) to generate new PUF keys in the same PUF label when the original key encounters duplicating risks. Once the modification is finished, the PUF label will be fixed without any changes. Therefore, we believe there is no conflict in combining machine vision, artificial intelligence, and machine learning techniques with any fixed diamond-PUF anti-counterfeiting labels for better performance.

To clarify this point, the above discussion about the improvement using advanced algorithms has been rearranged in the newly added Discussion section in the revised manuscript, see page 7, line 303.

Q8 The transformation of capture to binary encoding image is the basis for calculation. Since digital key extraction and performance analysis were conducted by MATLAB, the codes should be offered for the work to be reproduced.

Response:

We thank the reviewer for pointing this out. We have uploaded our MATLAB codes together with our source data of Fig. 3a(ii) and Fig. 4b as a Source Data file upon the submission of our revision files.

Q9 The “multimodal” and “dynamic” are key information and contribution, which can be highlighted and added in the Title.

Response:

We thank the reviewer’s suggestion. We have revised the Title accordingly: “Multimodal, dynamic, and unclonable anti-counterfeiting using robust diamond microparticles on heterogeneous substrate”

Other Changes made in the Revised Manuscript

-Page 1, line, 1, the title was revised to: “Multimodal, dynamic, and unclonable anti-counterfeiting using robust diamond microparticles on heterogeneous substrate”

-Page 1, line, 10, the affiliation of “Jing Wang” and “Lei Shao” was changed to: “State Key Laboratory of Optoelectronic Materials and Technologies, Guangdong Province Key Laboratory of Display Material and Technology, School of Electronics and Information Technology, Sun Yat-sen University, Guangzhou, China”

-Page 1, line, 16, the affiliation of 5 was changed to: “Primemax Biotech Limited”

-Page 1, line, 27, to meet the formatting requirement, the Abstract was slightly revised to 150 words.

-Page 2, line, 56, “nanoparticles (NPs)” was changed to “materials”.

-Page 2, line, 63, “organic” was changed to “polymer”.

-Page 2, line, 82, “chemical” was changed to “surface”.

-Page 2, line, 83, “focused ion beam” was changed to “reactive ion”.

-Page 2, line, 97, “Compared with other most studied optical PUFs (e.g., plasmonic NPs⁷, organic dots¹², quantum dots¹⁴, photonic crystals¹⁸, perovskite NPs²⁷, and upconversion NPs³¹),” was deleted.

-Page 5, line 208, “a” was deleted in the sentence “exhibit significant SiV signals”.

-Page 6, line, 259, “In this way, a kind of dynamic information storage, encryption and transformation material can be successfully developed.” was deleted.

-Page 6, line 259, “Therefore”, “diamond-PUF” and “for a higher level of security” were added to the last sentence.

-Page 7, line, 310, “In this paper” was changed to “In summary”.

-In Fig. 1c, we apologize for our mistake in calculating C (4) in Fig. 1c, and have replaced it with the correct one: $C(4) = (n + 1) \times P^m$.

-To meet the formatting requirement, we have provided the figures of the manuscript in individual files.

-To meet the formatting requirement, we have placed the “Acknowledgements”, Author Contributions”, “Competing Interests”, and “Figure Legends” sections after the “References” section.

References

- R1. Liu, K., *et al.* Tailoring of Typical Color Centers in Diamond for Photonics. *Adv. Mater.*, e2000891 (2020).
- R2. Trojanek, F., *et al.* Light emission dynamics of silicon vacancy centers in a polycrystalline diamond thin film. *Nanoscale*, (2023).
- R3. Auciello, O. & Aslam, D. M. Review on advances in microcrystalline, nanocrystalline and ultrananocrystalline diamond films-based micro/nano-electromechanical systems technologies. *J. Mater. Sci.* **56**, 7171–7230 (2021).
- R4. Ondic, L., *et al.* Photonic crystal cavity-enhanced emission from silicon vacancy centers in polycrystalline diamond achieved without postfabrication fine-tuning. *Nanoscale* **12**, 13055-13063 (2020).
- R5. Arnault, J.-C., Saada, S. & Ralchenko, V. Chemical Vapor Deposition Single-Crystal Diamond: A Review. *Phys. Status Solidi RRL* **16**, (2021).
- R6. Tzeng, Y. K., *et al.* Vertical-Substrate MPCVD Epitaxial Nanodiamond Growth. *Nano Lett.* **17**, 1489–1495 (2017).
- R7. Malik, R. J. *III-V Semiconductor Materials and devices*. Elsevier (2012).
- R8. Gupta, K., Gupta, N., Gupta, K. & Gupta, N. *Recent Advances in Semiconducting Materials and Devices*. Springer (2016).
- R9. Schmidt, I. & Benndorf, C. Low temperature CVD diamond deposition using halogenated precursors — deposition on low melting materials: Al, Zn and glass. *Diamond Relat. Mater.* **10**, 347-351 (2001).
- R10. Xiao, X., Birrell, J., Gerbi, J. E., Auciello, O. & Carlisle, J. A. Low temperature growth of ultrananocrystalline diamond. *J. Appl. Phys.* **96**, 2232-2239 (2004).
- R11. Das, D. & Roy, A. Growth of Nanostructured Diamond Films on Glass Substrates by Low-Temperature Microwave Plasma-Enhanced Chemical Vapor Deposition for Applications in Nanotribology. *ACS Appl. Nano Mater.* **5**, 3558-3571 (2022).
- R12. Piazza, F. & Morell, G. Synthesis of diamond at sub 300 °C substrate temperature. *Diamond Relat. Mater.* **16**, 1950-1957 (2007).
- R13. Nunn, N., Torelli, M., McGuire, G. & Shenderova, O. Nanodiamond: A High Impact Nanomaterial. *Curr. Opin. Solid State Mater. Sci.* **21**, 1–9 (2017).
- R14. Tinwala, H. & Wairkar, S. Production, Surface Modification and Biomedical Applications of Nanodiamonds: A Sparkling Tool for Theranostics. *Mater. Sci. Eng. C* **97**, 913–931 (2019).
- R15. Yiu, Y. C. & Chu, Z. A multilevel optical anti-counterfeiting system based on color space correlated Raman spectroscopy of diamond. *Adv. Photonics Res.*, In press (2023).
- R16. Bradac, C., Gao, W., Forneris, J., Trusheim, M. E. & Aharonovich, I. Quantum nanophotonics with group IV defects in diamond. *Nat. Commun.* **10**, 5625 (2019).
- R17. Zhang, T., *et al.* High-quality diamond microparticles containing SiV centers grown by chemical vapor deposition with preselected seeds. *J. Mater. Chem. C* **10**, 13734–13740 (2022).
- R18. Neu, E., *et al.* Narrowband Fluorescent Nanodiamonds Produced from Chemical Vapor Deposition Films. *Appl. Phys. Lett.* **98**, 243107 (2011).
- R19. Zhang, T., *et al.* Toward Quantitative Bio-sensing with Nitrogen–Vacancy Center in Diamond. *ACS Sens.* **6**, 2077–2107 (2021).

- R20. Neu, E., Agio, M. & Becher, C. Photophysics of single silicon vacancy centers in diamond: implications for single photon emission. *Opt. Express* **20**, 19956-19971 (2012).
- R21. Kaupp, H., *et al.* Purcell-Enhanced Single-Photon Emission from Nitrogen-Vacancy Centers Coupled to a Tunable Microcavity. *Phys. Rev. Appl.* **6**, (2016).
- R22. Iwasaki, T., *et al.* Germanium-Vacancy Single Color Centers in Diamond. *Sci. Rep.* **5**, 12882 (2015).
- R23. Tchernij, S. D., *et al.* Single-Photon-Emitting Optical Centers in Diamond Fabricated upon Sn Implantation. *ACS Photonics* **4**, 2580-2586 (2017).
- R24. Ditalia Tchernij, S., *et al.* Single-Photon Emitters in Lead-Implanted Single-Crystal Diamond. *ACS Photonics* **5**, 4864-4871 (2018).
- R25. Aharonovich, I., *et al.* Enhanced single-photon emission in the near infrared from a diamond color center. *Phys. Rev. B* **79**, (2009).
- R26. Aharonovich, I., *et al.* Chromium single-photon emitters in diamond fabricated by ion implantation. *Phys. Rev. B* **81**, (2010).
- R27. Zaitsev, A. M. Vibronic spectra of impurity-related optical centers in diamond. *Phys. Rev. B* **61**, 12909-12922 (2000).
- R28. Neu, E., *et al.* Single Photon Emission from Silicon-Vacancy Colour Centres in Chemical Vapour Deposition Nano-Diamonds on Iridium. *New J. Phys.* **13**, 025012 (2011).
- R29. Malykhin, S., Mindarava, Y., Ismagilov, R., Jelezko, F. & Obraztsov, A. Control of NV, SiV and GeV Centers Formation in Single Crystal Diamond Needles. *Diamond Relat. Mater.* **125**, 109007 (2022).
- R30. Westerhausen, M. T., *et al.* Controlled Doping of GeV and SnV Color Centers in Diamond Using Chemical Vapor Deposition. *ACS Appl. Mater. Interfaces* **12**, 29700–29705 (2020).
- R31. Wu, E., *et al.* Room temperature triggered single-photon source in the near infrared. *New J. Phys.* **9**, 434-434 (2007).
- R32. Ren, W., Lin, G., Clarke, C., Zhou, J. & Jin, D. Optical Nanomaterials and Enabling Technologies for High-Security-Level Anticounterfeiting. *Adv. Mater.* **32**, 1901430 (2020).
- R33. Huo, Y., Yang, Z., Wilson, T. & Jiang, C. Recent Progress in SERS-Based Anti-counterfeit Labels. *Adv. Funct. Mater.* **9**, 2200201 (2022).
- R34. Arppe, R. & Sorensen, T. J. Physical unclonable functions generated through chemical methods for anti-counterfeiting. *Nat. Rev. Chem.* **1**, 0031 (2017).
- R35. Yu, X., Zhang, H. & Yu, J. Luminescence anti-counterfeiting: From elementary to advanced. *Aggregate* **2**, 20-34 (2021).
- R36. Ibrar, M. & Skrabalak, S. E. Designer Plasmonic Nanostructures for Unclonable Anticounterfeit Tags. *Small Struct.* **2**, (2021).
- R37. Suo, H., Zhu, Q., Zhang, X., Chen, B., Chen, J. & Wang, F. High-security anti-counterfeiting through upconversion luminescence. *Mater. Today Phys.* **21**, (2021).
- R38. Abdollahi, A., Roghani-Mamaqani, H., Razavi, B. & Salami-Kalajahi, M. Photoluminescent and Chromic Nanomaterials for Anticounterfeiting Technologies: Recent Advances and Future Challenges. *ACS Nano* **14**, 14417-14492 (2020).

- R39. Mochalin, V. N., Shenderova, O., Ho, D. & Gogotsi, Y. The Properties and Applications of Nanodiamonds. *Nat. Nanotechnol.* **7**, 11–23 (2011).
- R40. Zarska, M., *et al.* Biological safety and tissue distribution of (16-mercaptophexadecyl)trimethylammonium bromide-modified cationic gold nanorods. *Biomaterials* **154**, 275-290 (2018).
- R41. Smith, J. D., *et al.* Plasmonic Anticounterfeit Tags with High Encoding Capacity Rapidly Authenticated with Deep Machine Learning. *ACS Nano* **15**, 2901-2910 (2021).
- R42. Alkilany, A. M. & Murphy, C. J. Toxicity and cellular uptake of gold nanoparticles: what we have learned so far? *J. Nanopart. Res.* **12**, 2313-2333 (2010).
- R43. Zhang, Y., Gu, Y., He, J., Thackray, B. D. & Ye, J. Ultrabright gap-enhanced Raman tags for high-speed bioimaging. *Nat. Commun.* **10**, 3905 (2019).
- R44. Gu, Y., He, C., Zhang, Y., Lin, L., Thackray, B. D. & Ye, J. Gap-enhanced Raman tags for physically unclonable anticounterfeiting labels. *Nat. Commun.* **11**, 516 (2020).
- R45. Kayaci, N., Ozdemir, R., Kalay, M., Kiremitler, N. B., Usta, H. & Onses, M. S. Organic Light-Emitting Physically Unclonable Functions. *Adv. Funct. Mater.*, 2108675 (2021).
- R46. Liu, Y., *et al.* Inkjet-printed unclonable quantum dot fluorescent anti-counterfeiting labels with artificial intelligence authentication. *Nat. Commun.* **10**, 2409 (2019).
- R47. Wu, J., *et al.* Unclonable Photonic Crystal Hydrogels with Controllable Encoding Capacity for Anticounterfeiting. *ACS Appl. Mater. Interfaces* **14**, 2369-2380 (2022).
- R48. Yakunin, S., *et al.* Radiative lifetime-encoded unicolour security tags using perovskite nanocrystals. *Nat. Commun.* **12**, 981 (2021).
- R49. Liu, X., *et al.* Binary temporal upconversion codes of Mn²⁺-activated nanoparticles for multilevel anti-counterfeiting. *Nat. Commun.* **8**, 899 (2017).
- R50. <https://m.tb.cn/h.Un9CW7A?tk=LhJedhG45Ge>.
- R51. Lee, S. K., Kim, J. H., Jeong, M. G., Song, M. J. & Lim, D. S. Direct deposition of patterned nanocrystalline CVD diamond using an electrostatic self-assembly method with nanodiamond particles. *Nanotechnology* **21**, 505302 (2010).
- R52. Andrich, P., Li, J., Liu, X., Heremans, F. J., Nealey, P. F. & Awschalom, D. D. Microscale-Resolution Thermal Mapping Using a Flexible Platform of Patterned Quantum Sensors. *Nano Lett.* **18**, 4684–4690 (2018).
- R53. Xu, Z., *et al.* On-Demand, Direct Printing of Nanodiamonds at the Quantum Level. *Adv. Sci.*, e2103598 (2021).
- R54. Taylor, A. C., Edgington, R. & Jackman, R. B. Patterning of nanodiamond tracks and nanocrystalline diamond films using a micropipette for additive direct-write processing. *ACS Appl. Mater. Interfaces* **7**, 6490–6495 (2015).
- R55. Chen, Y.-C., Tzeng, Y., Cheng, A.-J., Dean, R., Park, M. & Wilamowski, B. M. Inkjet printing of nanodiamond suspensions in ethylene glycol for CVD growth of patterned diamond structures and practical applications. *Diamond Relat. Mater.* **18**, 146-150 (2009).
- R56. Jiang, M., Kurvits, J. A., Lu, Y., Nurmikko, A. V. & Zia, R. Reusable Inorganic Templates for Electrostatic Self-Assembly of Individual Quantum Dots, Nanodiamonds, and Lanthanide-Doped Nanoparticles. *Nano Lett.* **15**, 5010-5016 (2015).
- R57. Rius, G., Baldi, A., Ziaie, B. & Atashbar, M. Z. Introduction to micro-/nanofabrication. *Springer Handbook of Nanotechnology*, 51-86 (2017).

- R58. https://www.bain.com/contentassets/a53a9fa8bf5247a3b7bb0b10561510c2/bain_diamond_report_2018.pdf.
- R59. https://pdf.dfcfw.com/pdf/H3_AP202201171540960153_1.pdf.
- R60. Wang, H.-J. & Yang, T. A review on laser drilling and cutting of silicon. *J. Eur. Ceram. Soc.* **41**, 4997-5015 (2021).
- R61. Moreau, W. M. *Semiconductor lithography: principles, practices, and materials*. Springer Science & Business Media (2012).
- R62. Köhler, M. *Etching in microsystem technology*. John Wiley & Sons (2008).
- R63. Singh, S., Thomas, V., Martyshkin, D., Kozlovskaya, V., Kharlampieva, E. & Catledge, S. A. Spatially controlled fabrication of a bright fluorescent nanodiamond-array with enhanced far-red Si-V luminescence. *Nanotechnology* **25**, 045302 (2014).
- R64. Lauten, F. S., Shigesato, Y. & Sheldon, B. W. Diamond nucleation on unscratched SiO₂ substrates. *Appl. Phys. Lett.* **65**, 210-212 (1994).
- R65. Haque, A., Gupta, S. & Narayan, J. Characteristics of Diamond Deposition on Al₂O₃, Diamond-like Carbon, and Q-Carbon. *ACS Appl. Electron. Mater.* **2**, 1323-1334 (2020).
- R66. May, P. W., Tsai, H. Y., Wang, W. N. & Smith, J. A. Deposition of CVD diamond onto GaN. *Diamond Relat. Mater.* **15**, 526-530 (2006).
- R67. Soleimanzadeh, R., Naamoun, M., Floriduz, A., Khadar, R. A., van Erp, R. & Matioli, E. Seed Dribbling Method for the Growth of High-Quality Diamond on GaN. *ACS Appl. Mater. Interfaces* **13**, 43516-43523 (2021).
- R68. Rycewicz, M., *et al.* Low-strain sensor based on the flexible boron-doped diamond-polymer structures. *Carbon* **173**, 832-841 (2021).
- R69. Xie, Y., *et al.* Diamond thin films integrated with flexible substrates and their physical, chemical and biological characteristics. *J. Phys. D: Appl. Phys.* **54**, 384004 (2021).

REVIEWERS' COMMENTS

Reviewer #1 (Remarks to the Author):

The authors have answered the queries raised in the last round of review however I feel that it still does not classify as a breakthrough.

1. It is wrong to claim that high-temperature is favoured in electronics industry. What is favoured is the bare minimum to get high-quality material not the temperature. If the temperature can be lowered it will be done because that brings down cost and 900 degC is detrimental to many electronic industry components.

2. The process used in electronic industry CVD or other wise is very different from diamond growth, so the same system cannot be used.

3. Another publication in the past(<https://doi.org/10.1002/adfm.202102108>) has shown the use of diamond for unclonable labels, so this is again nothing new.

In summary, the work is interesting but not upto the standards needed for Nature Comms.

Reviewer #2 (Remarks to the Author):

I am okay with the revised version since the authors responded all of my concerns.

Reviewer #3 (Remarks to the Author):

I think that the authors have fully addressed the concerns and the quality of the manuscript is highly enhanced.

POINT-BY-POINT REPLY TO REVIEWER COMMENTS

We thank all the reviewers for their valuable comments. Please find below the point-by-point reply to the comments, with the reply in blue.

REVIEWERS' COMMENTS

Reviewer #1 (Remarks to the Author):

The authors have answered the queries raised in the last round of review however I feel that it still does not classify as a breakthrough.

1. It is wrong to claim that high-temperature is favoured in electronics industry. What is favoured is the bare minimum to get high-quality material not the temperature. If the temperature can be lowered it will be done because that brings down cost and 900 degC is detrimental to many electronic industry components.

Response:

We thank the reviewer for acknowledging that we have answered the queries raised in the last round, and feel the reviewer might have misunderstood our previous response “*a relatively high-temperature growth condition is generally favored in the modern large-scale fabrication of high-quality semiconductor materials, because it normally ensures the great stability of materials*”. It is a generally accepted fact that high-temperature processes may be needed in electronics and optoelectronics industry, e.g., the growth and fabrication of very famous III-V and II-VI compound semiconductors^{R1-3}. In addition, several high-temperature steps (e.g., rapid thermal treatment and annealing) are also involved in the front end of line (FEOL) of typical semiconductor manufacturing^{R4-6}. Notice that we never claim that we will grow diamond-PUF together with those fabricated products, e.g., electronic devices which potentially will be damaged by CVD growth processes (e.g., high temperature). Instead, one can simply prefer to use the pre-fabricated diamond-PUF labels on the manufactured products operating at room temperature.

- R1. Morkoç, H., Strite, S., Gao, G. B., Lin, M. E., Sverdlov, B. & Burns, M. Large-band-gap SiC, III-V nitride, and II-VI ZnSe-based semiconductor device technologies. *J. Appl. Phys.* **76**, 1363–1398 (1994).
- R2. Jones, A. C. & O'Brien, P. *CVD of Compound Semiconductors: Precursor Synthesis, Development and Applications*. John Wiley & Sons (2008).

- R3. Razeghi, M. *The MOCVD Challenge: A survey of GaInAsP-InP and GaInAsP-GaAs for photonic and electronic device applications, Second Edition*. CRC Press. (2010).
- R4. Lin, L., Peng, H. & Liu, Z. Synthesis challenges for graphene industry. *Nat. Mater.* **18**, 520–524 (2019).
- R5. Nishi, Y. & Doering, R. *Handbook of Semiconductor Manufacturing Technology, Second Edition*. CRC press (2008).
- R6. Van Zant, P. *Microchip fabrication, Sixth Edition*. McGraw-Hill Education (2014).

2. The process used in electronic industry CVD or other wise is very different from diamond growth, so the same system cannot be used.

Response:

We feel the reviewer might have misunderstood us. We do not intend to use common CVD used in electronic industry to grow diamond material. They have completely different process objectives and reaction mechanisms. Actually, until now, various kinds of CVD techniques have been developed for the growth of diamond, such as hot filament CVD (HFCVD), microwave plasma CVD (MPCVD), radio-frequency plasma-enhanced CVD (RFCVD), DC arc plasma jet CVD, combustion flame CVD, etc.^{R7-10}. And some of them (especially MPCVD used in this work) have already achieved the capability of **mass commercialization** of diamond products for electronic, thermal, optical, and quantum technology^{R9-11}. In addition, as one of the most promising third-generation semiconductor materials, significant achievements have been made in past years, i.e., various electronic and photonic devices based on the CVD-grown diamond^{R12-14}.

- R7. Balmer, R. S., *et al.* Chemical vapour deposition synthetic diamond: materials, technology and applications. *J. Phys.: Condens. Matter.* **21**, 364221 (2009).
- R8. Gracio, J. J., Fan, Q. H. & Madaleno, J. C. Diamond growth by chemical vapour deposition. *J. Phys. D: Appl. Phys.* **43**, 374017 (2010).
- R9. Yan, C.-s., Vohra, Y. K., Mao, H.-k. & Hemley, R. J. Very high growth rate chemical vapor deposition of single-crystal diamond. *Proc. Natl. Acad. Sci. U. S. A.* **99**, 12523–12525 (2002).
- R10. Lu, F. X. Past, present, and the future of the research and commercialization of CVD diamond in China. *Functional Diamond* **2**, 119–141 (2022).
- R11. Element Six. <https://e6cvd.com/us/>.
- R12. Sussmann, R. S. *CVD diamond for electronic devices and sensors*. John Wiley & Sons (2009).
- R13. Aharonovich, I., Greentree, A. D. & Prawer, S. Diamond photonics. *Nat. Photonics* **5**, 397–405 (2011).
- R14. Dang, C., *et al.* Achieving Large uniform tensile elasticity in microfabricated diamond. *Science* **371**, 76–78 (2021).

3. Another publication in the past(<https://doi.org/10.1002/adfm.202102108>) has shown the use of diamond for unclonable labels, so this is again nothing new.

In summary, the work is interesting but not up to the standards needed for Nature Comms.

Response:

We thank the reviewer's comment. The novelty of our work is significantly different from the mentioned paper, which mainly focused on utilizing Raman signal of biocompatible HPHT microdiamonds mixed in biocompatible silk film for binary encoding. Our work demonstrated a robust, dynamic, trustworthy, and intelligent anti-counterfeiting strategy based on the diamond microparticles with SiV centers on Si wafer by one-step heterogeneous CVD fabrication, with the scalable commercialization ability. The PUF nature of the light scattering patterns, spectra, shape, and SiV PL signals of the diamond microparticles offer multimodal encoding capability. Moreover, time-dependent encoding could be achieved by dynamically modulating the diamond-PUF labels via post air oxidation. Additionally, the developed labels exhibited ultrahigh stability in extreme application scenarios, including harsh chemical environments, high temperature, mechanical forces, and UV light irradiation.

In addition, we feel the work mentioned by the reviewer may serve as a proper reference for emphasizing the biosafety of using diamond material for anti-counterfeiting applications, therefore, we have cited it as ref 2 in our Supplementary Table 1 of Supplementary Information.

Reviewer #2 (Remarks to the Author):

I am okay with the revised version since the authors responded all of my concerns.

Response:

We sincerely thank the reviewer for the kind comment and satisfying our previous response and revision.

Reviewer #3 (Remarks to the Author):

I think that the authors have fully addressed the concerns and the quality of the manuscript is highly enhanced.

Response:

We sincerely thank the reviewer for the kind comment and satisfying our previous response and revision.